# QuantDemoire: Quantization with Outlier Aware for Image Demoiréing

## Abstract

Demoiréing aims to remove moiré artifacts that often occur in images. While recent deep learning-based methods have achieved promising results, they typically require substantial computational resources, limiting their deployment on edge devices. Model quantization offers a compelling solution. However, directly applying existing quantization methods to demoiréing models introduces severe performance degradation. The main reasons are distribution outliers and weakened representations in smooth regions. To address these issues, we propose QuantDemoire, a post-training quantization framework tailored to demoiréing. It contains two key components. **First**, we introduce an outlier-aware quantizer to reduce errors from outliers. It uses sampling-based range estimation to reduce activation outliers, and keeps a few extreme weights in FP16 with negligible cost. **Second**, we design a frequency-aware calibration strategy. It emphasizes low- and mid-frequency components during fine-tuning, which mitigates banding artifacts caused by low-bit quantization. Extensive experiments validate that our QuantDemoire achieves large reductions in parameters and computation while maintaining quality. Meanwhile, it outperforms existing quantization methods by over **4 dB** on W4A4.

## 1 Introduction

Image demoiréing is a low-level vision task that aims to remove colored stripes caused by frequency aliasing between display screens and imaging sensors. These artifacts degrade the perceptual quality of captured images and interfere with subsequent visual analysis and recognition. Due to the complex and diverse forms of moiré patterns, achieving effective demoiréing remains a highly challenging task (Sidorov & Kokaram, 2002; Liu et al., 2015; Wang et al., 2023).

In recent years, with the rapid progress of deep learning, neural networks have achieved remarkable performance in demoiréing (Cheng et al., 2019; Zheng et al., 2020; Yu et al., 2022; Xu et al., 2023; Liu et al., 2025). These models can adapt to the diversity of moiré patterns, achieving excellent performance. However, they usually involve millions of parameters and heavy computation. This makes them unsuitable for devices with extremely limited computing power, such as drones, portable cameras and IoT sensors. (Cheng et al., 2019; Zheng et al., 2020). These devices are also the most important application scenarios of image demoiréing. Thus, achieving efficient demoiréing while maintaining high quality is critical for practical usage.

Model quantization (Nagel et al., 2021) offers an effective solution to this challenge. It compresses weights and activations from 32-bit floating-point to low-bit (*e.g.*, 2∼8 bit) integers. This can significantly reduce storage overhead while converting costly floating-point operations into efficient integer operations. Therefore, the quantization can accelerate inference and lower power consumption, making it well-suited for resource-constrained devices and hardware accelerators. Currently, quantization is widely applied to vision tasks (Esser et al., 2019; Li et al., 2023; Qin et al., 2023).

However, in the demoiréing task, there is a lack of corresponding quantization methods. To accommodate deployment in scenarios with extremely limited computing power, it is imperative to compress demoiréing models to ultra-low bit-widths (e.g., 4-bit). As a result, directly applying existing quantization (Jacob et al., 2018; Li et al., 2024) results in severe performance loss. The degradation arises primarily from two factors. **First**, outliers in weights and activations expand the quantization range. This reduces the precision of most effective values and weakens the demoiréing performance. **Second**, quantization weakens representations in smooth regions. It introduces banding artifacts in mid- and low-frequency components.

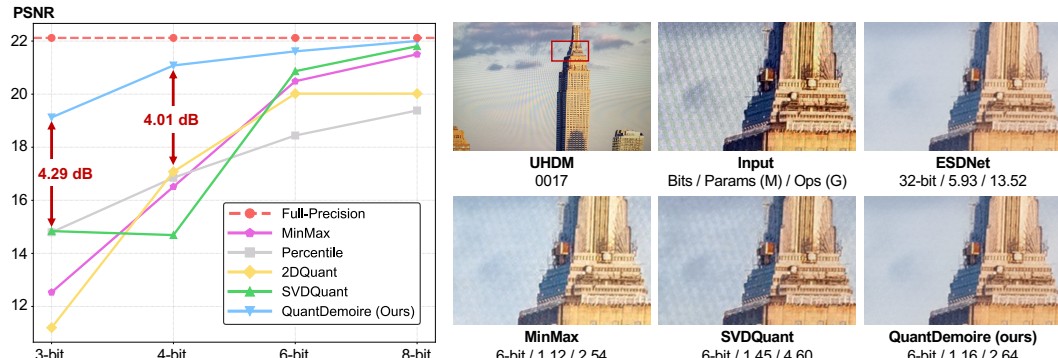

Figure 1: Comparison with recent quantization methods on UHDM (Yu et al., 2022). The full-precision backbone is ESDNet (Yu et al., 2022). Left: PSNR performance at different bit-widths. Right: visual comparison. Our QuantDemoire demonstrates superior efficiency and performance.

To address the above issues, we propose QuantDemoire, a post-training quantization (PTQ) framework tailored for image demoiréing tasks. Our core idea is to mitigate the accuracy loss from two perspectives: the quantizer and the calibration. **(1) Quantizer.** We design the outlier-aware quantizer to reduce the impact of outliers. It uses random sampling to estimate and remove activation outliers. It also keeps a very small fraction of extreme weights in FP16, with negligible overhead. **(2) Calibration.** We introduce the frequency-aware calibration strategy. It explicitly emphasizes mid- and low-frequency components during the optimization of quantization boundaries. This alleviates the banding artifacts in smooth regions caused by quantization.

We conduct extensive experiments to demonstrate that our QuantDemoire can achieve significant efficiency improvements while maintaining strong demoiréing performance. It consistently outperforms existing quantization methods across datasets and bit-widths. As shown in Fig. 1, compared with the full-precision model, our method reduces parameters and computation by over **86.6%** at 4-bit with less than **4.7%** (PSNR) performance drop. Besides, QuantDemoire surpasses current SOTA quantization methods by more than **4 dB** (PSNR, 4-bit) on the UHDM (Yu et al., 2022) dataset.

Our main contributions are summarized as follows:

- We propose QuantDemoire, the first quantization framework tailored to image demoiréing. It reduces model overhead while preserving strong restoration ability.
- We design the outlier-aware quantizer that mitigates quantization errors by handling outliers in both activations and weights with random sampling.
- We propose the frequency-aware calibration strategy to optimize mid- and low-frequency features, alleviating quantization-induced banding artifacts.
- We validate our method on multiple datasets and bit-widths. QuantDemoire consistently outperforms existing quantization methods quantitatively and visually.

## 2 RELATED WORK

### 2.1 IMAGE DEMOIRÉING.

When photographing content displayed on a digital screen, the captured image often exhibits colored stripes known as moiré patterns. These patterns can substantially reduce the overall quality of the image. Early approaches to moiré pattern removal primarily relied on traditional mathematical methods, such as matrix decomposition (Liu et al., 2015) and spectral models (Sidorov & Kokaram, 2002). These techniques generally perform poorly when dealing with diverse types of moiré patterns. In recent years, with the rapid development of deep learning, methods (Cheng et al., 2019; He et al., 2019; Zheng et al., 2020; He et al., 2020; Yu et al., 2022; Xu et al., 2023; Xiao et al., 2024; Liu et al., 2025), based on deep neural networks, have been proposed. MopNet (He et al., 2019) utilizes multi-scale feature aggregation and attribute-aware classifiers to handle complex frequency. To further improve performance, FHDe$^2$Net (He et al., 2020) employs a cascaded architecture to remove multi-scale moiré patterns. Besides, ESDNet (Yu et al., 2022) integrates a semantic-aligned, scale-aware module that further improves robustness to moiré pattern scale variations. Despite favorable moiré pattern removal, these methods involve millions of parameters. The heavy computational cost makes them unsuitable for deployment on edge devices.

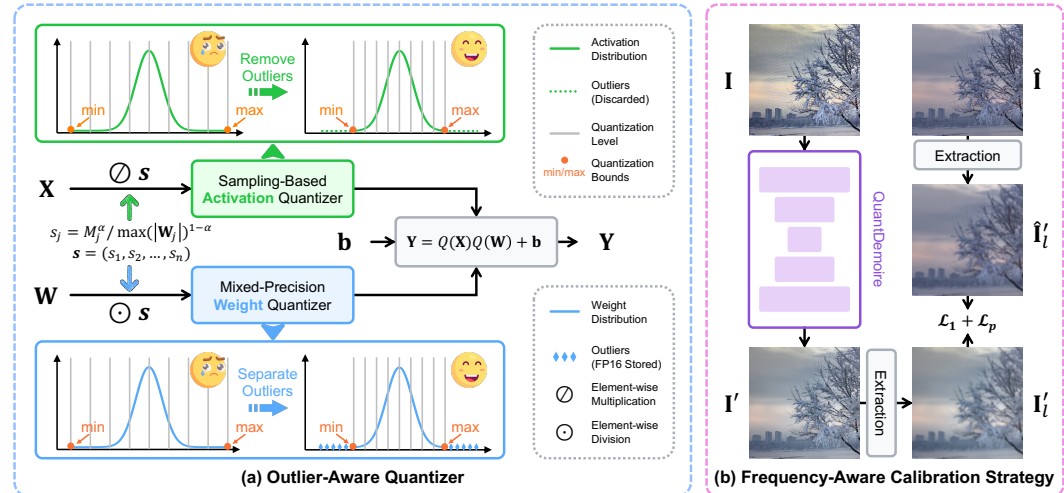

Figure 2: The overview of our QuantDemoire. (a) Outlier-Aware Quantizer: activations discard outliers through sample-based estimation, while weights preserve outliers in FP16 precision. (b) Frequency-Aware Calibration Strategy: quantizer parameters are optimized on mid- and low-frequency components, which are extracted through the frequency extraction process.

## 2.2 MODEL QUANTIZATION.

Quantization is widely used in neural network compression. By transforming weights and activations from 32-bit floating-point to low-bit integer formats, it effectively reduces both memory consumption and computational overhead. Existing quantization algorithms are generally categorized into quantization-aware training (QAT) (Esser et al., 2019; Chen et al., 2024; Qin et al., 2023) and post-training quantization (PTQ) (Liu et al., 2021; Hubara et al., 2021; Shang et al., 2023). LSQ (Esser et al., 2019) uses a learned step size to improve the performance of low-bit quantization. Although QAT achieves competitive performance (Nagel et al., 2021), Its high training cost limits its practical application. In contrast, PTQ is regarded as a lightweight and efficient approach, as it requires neither retraining nor labeled data. The MinMax strategy (Jacob et al., 2018) estimates quantization parameters directly from the global extrema of data distributions. SmoothQuant (Xiao et al., 2023) alleviates this issue by redistributing the quantization difficulty from activations to weights. SVDQuant (Li et al., 2024) applies singular value decomposition (SVD) to decompose the weight tensor into a low-rank component and a residual term, jointly capturing and mitigating outlier effects during quantization. Nevertheless, existing quantization methods suffer from severe performance degradation on demoiréing models. Specifically, due to the influence of outliers (Dettmers et al., 2022) and the reduced information density caused by quantization, the output image quality of quantized models deteriorates significantly, manifesting as color banding and poor moiré removal performance.

## 3 METHOD

In this section, we present our quantization framework, QuantDemoire, for efficient demoiréing (see Fig. 2). We introduce two key components. The first is the outlier-aware quantizer, which reduces the quantization error introduced by outliers in activations and weights. The second is the frequency-aware calibration strategy, which selectively emphasizes low- and mid-frequency components to mitigate the banding artifacts during the calibration phase for quantization boundaries.

## 3.1 PRELIMINARIES: QUANTIZATION

We first introduce model quantization (Jacob et al., 2018), which is employed to simulate quantization errors. Given a weight or activation tensor value $v$ to be fake-quantized and a quantization bit-width $b$, the processing procedure can be formally expressed as:

$$\Delta = \frac{u - l}{2^b - 1}, \quad z = \text{clip}(\text{Round}(-\frac{l}{\Delta}), 0, 2^b - 1),$$

$$v_z = \text{clip}(\text{Round}(\frac{v}{\Delta}) + z, 0, 2^b - 1), \tag{1}$$

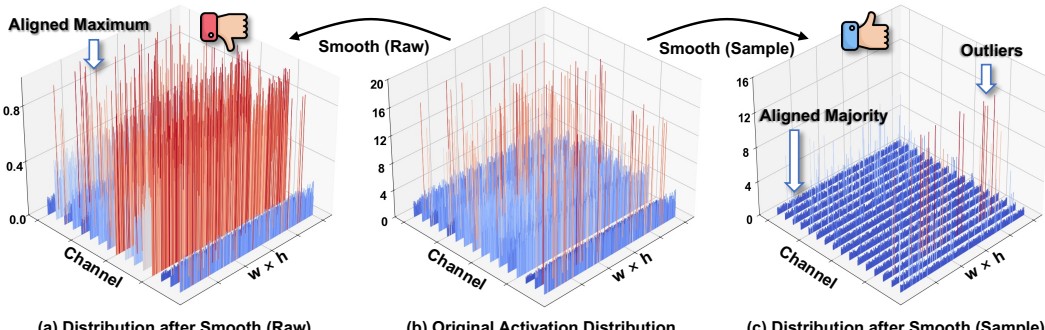

Figure 3: Activation distributions (Original, Smooth (Raw), Smooth (Sample)). (a) Original: ranges differ across channels; (b) Smooth (Raw): align maxima only; (c) Smooth (Sample): align the main body (excluding outliers), improving suitability for quantization.

where $l$ and $u$ denote the lower and upper quantization bounds, $\Delta$ is the scaling factor, $z$ is the zero-point offset, and $v_z$ is the integer-coded value of $v$. The clipping function is defined as $\text{clip}(v, l, u) = \max(\min(v, u), l)$, while $\text{Round}(\cdot)$ maps an input to its nearest integer. The corresponding dequantization process can be formulated as:

$$Q(v) = \Delta \times (v_z - z), \tag{2}$$

where $Q(\cdot)$ denotes the operation of fake quantization. For a quantized convolutional layer, given the input activation $\mathbf{X}$, weight $\mathbf{W}$, and bias $\mathbf{b}$, the output $\mathbf{Y}$ can be obtained through:

$$\mathbf{Y} = Q(\mathbf{X}) \cdot Q(\mathbf{W}) + \mathbf{b}. \tag{3}$$

Because $\text{Round}(\cdot)$ is non-differentiable, following prior works (Qin et al., 2023; Li et al., 2024), the straight-through estimator (STE) is adopted to approximate its gradient:

$$\frac{\partial Q(v)}{\partial v} \approx \begin{cases} 1 & \text{if } v \in [l, u], \\ 0 & \text{otherwise.} \end{cases} \tag{4}$$

Moreover, SmoothQuant (Xiao et al., 2023) is widely adopted in quantization to address the challenge arising from the variation in maximum magnitudes across channels. It applies a per-channel scaling transformation to redistribute the quantization difficulty from activations to weights. Given an activation $\mathbf{X}$ and a weight tensor $\mathbf{W}$ for the $j$-th input channel, the smoothing factor $s_j$ is defined as:

$$s_j = \max(|\mathbf{X}_j|)^\alpha / \max(|\mathbf{W}_j|)^{1-\alpha},$$
$$\mathbf{s} = (s_1, s_2, \ldots, s_n), \tag{5}$$

where $\alpha$ is a balancing hyperparameter controlling the trade-off between activation and weight quantization, $n$ denotes the number of input channels, and $\mathbf{s}$ is the smoothing coefficient vector.

Despite these developments, existing quantization methods still lead to substantial performance degradation when applied to demoiréing models. Therefore, we propose the outlier-aware quantizer and the frequency-aware calibration strategy to enhance the performance of demoiréing.

## 3.2 Outlier-aware quantizer

**Challenges.** Despite the progress in neural network quantization, existing methods still face critical limitations, particularly due to the presence of outliers. They can lead to two key problems: *quantization range expansion* and *inaccurate smoothing factor estimation*.

*Challenge I. Quantization range expansion.* Outliers enlarge the quantization range either within a single channel or across the entire tensor, thereby reducing information density and, as a result, further aggravating quantization error for normal values.

*Challenge II. Inaccurate smoothing factor estimation.* When the maximum absolute value, which is an outlier, is used to compute the smoothing factor, the maximum values across channels can be aligned. However, as shown in Fig. 3 (Smooth (Raw)), significant range disparities remain among the non-outlier values across channels due to the large difference between outliers and non-outliers.

Some existing post-training quantization methods (such as percentile (Li et al., 2019)) are specifically designed to address the problems caused by outliers. However, these methods lack a design specifically tailored to the outlier distributions observed in demoiréing models. Consequently, they fail to effectively mitigate the quantization errors introduced by these outliers.

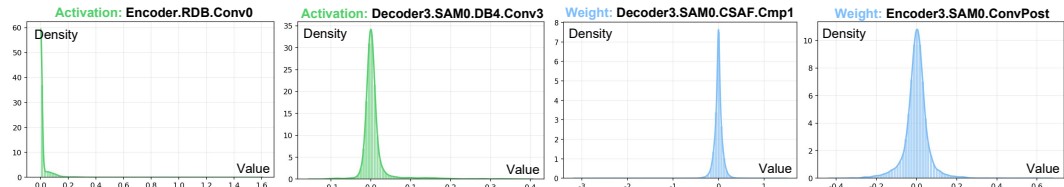

Figure 4: Visualization of **activation** and **weight** distributions from randomly selected layers in ESDNet (Yu et al., 2022). The distributions are approximately Gaussian or exponential.

**Sampling-Based Activation Quantizer.** To address the problems caused by outliers, we propose a random sampling-based method for estimating activation ranges. From Fig. 4, it can be observed that, most activations conform to either a Gaussian or an exponential distribution. Such distributions typically exhibit a small number of outliers that deviate from the normal range on one or both sides. More distribution visualizations are provided in the supplementary material to support this point.

To mitigate the impact of these outliers on accuracy, we adopt a strategy of randomly sampling a small subset of activation values in a channel or tensor (Fig. 2a). Sample$\gamma(\cdot)$ denotes a subset formed by randomly selecting a proportion $\gamma$ of elements from the activations. This approach reduces the probability of selecting outliers, thereby alleviating the quantization error, while at the same time ensuring an accurate estimation of the distribution range of the majority of non-outlier values.

In LLMs, outliers typically arise from the large magnitudes of certain activation channels (Xiao et al., 2023). This allows the smoothing factor to be calculated directly from the per-channel maximum absolute value. In contrast, demoiréing models, which are primarily composed of convolutional blocks, feature outliers that are mostly sporadic elements. Consequently, the presence of these elemental outliers within a channel can skew the estimation of its overall magnitude. Therefore, we estimate the maximum magnitude of activations for each channel by randomly sampling:

$$M_j = \max(|\text{Sample}_{\gamma_1}(\mathbf{X}_j)|), \tag{6}$$

where $\mathbf{X}_j$ is the $j$-th input channel of activation, $M_j$ is the estimated maximum magnitude and, $\gamma_1$ is the proportion of elements sampled from each activation channel. By restricting the estimation to a sampled subset, the proposed method effectively captures the main body of activations while reducing the influence of outlier values. For the calculation of the maximum weight magnitude, we directly take the largest absolute value. The computation of the smoothing factor can be written as:

$$s_j = M_j^\alpha / \max(|\mathbf{W}_j|)^{1-\alpha}. \tag{7}$$

Visualizations in Fig. 3 reveals that calculating smoothing factor with sampled maximum magnitude of activations (Smooth (Sample)) achieves better smoothing of non-outlier values (*i.e.*, the majority of the distribution). After being divided by the smoothing factor, the activation's per-channel magnitude have been aligned. However, the outlier elements still remain in the activation tensor. Therefore, in computing the quantization bound, we still employ a sampling approach, with a proportion $\gamma_2$:

$$u_a = \max(\text{Sample}_{\gamma_2}(\mathbf{X} \oslash \mathbf{s})),$$
$$l_a = \min(\text{Sample}_{\gamma_2}(\mathbf{X} \oslash \mathbf{s})), \tag{8}$$

where $l_a$, $u_a$ denote the lower and upper bounds of the quantizer. $\mathbf{X}$ is the activation tensor and $\mathbf{s}$ is the smoothing factor. Experiments show that computing the quantization bound via random sampling outperforms clipping activations at a fixed percentile in reducing quantization error for demoiréing models. A detailed mathematical analysis in supplementary materials substantiates this result.

Finally, the derived $l_a$ and $u_a$ are applied in Eq. (1) to perform activation quantization, forming the sampling-based activation quantizer. This quantizer is able to effectively mitigate the impact of outliers on the quantization accuracy, thereby improving model performance. To avoid extra inference-time overhead, the outlier-aware quantizer uses static quantization. We compute the smoothing factor and quantization bounds using a calibration set comprising 200 pairs of training dataset images.

**Mixed-Precision Weight Quantizer.** As illustrated in Fig. 4, similar to the behavior observed in activation distributions, the weight distribution of the network exhibits an approximately Gaussian distribution. Therefore, outliers of the weight exist on the left and right side of the distribution. A straightforward way to handle these extreme outlier values is to apply the same sampling-based estimation method proposed in activation quantization. However, unlike activations, weight tensors of convolutional modules generally contain far fewer elements. Consequently, the direct truncation of extreme weight values leads to a notable degradation in the performance of the quantized model.

One solution is to store outliers in FP16 via mixed precision. However, most existing quantization methods apply mixed precision at the layer or channel granularity (Feng et al., 2025). Because outliers in convolutional-module weights occur sporadically at random positions, these coarse-grained schemes are not directly applicable.

To overcome this limitation, we propose preserving the outliers in FP16 precision. The illustration is in Fig. 2a. By preserving a proportion $\beta$ of outliers at both ends, we can calculate the corresponding lower and upper percentile thresholds: $T_{\text{low}} = P_\beta(\mathbf{W})$ and $T_{\text{high}} = P_{1-\beta}(\mathbf{W})$. Here, $P_\beta(\mathbf{W})$ denotes the $\beta$-th percentile of $\mathbf{W}$. The complete weight quantization process can then be expressed:

$$\mathbf{W}_{\text{outlier}} = \{w_i \in \mathbf{W} \mid w_i < T_{\text{low}} \text{ or } w_i > T_{\text{high}}\},$$

$$\mathbf{W}_{\text{normal}} = \mathbf{W} \setminus \mathbf{W}_{\text{outlier}}, \quad \hat{\mathbf{W}} = \mathbf{W}_{\text{outlier}} \cup Q(\mathbf{W}_{\text{normal}}), \tag{9}$$

where $\mathbf{W}_{\text{outlier}}$ and $\mathbf{W}_{\text{normal}}$ denote the sets of outlier and non-outlier weights, respectively; $\hat{\mathbf{W}}$ represents the final quantized weight. Experimental results indicate that $\beta = 0.005$ is sufficient.

During inference, we decompose the convolution into two branches: a low-bit branch and an FP16 branch. The low-bit branch performs the convolution using quantized weights and quantized activations. The FP16 branch computes, on an element-wise basis, the contributions of stored weight outliers to the output. The results of the two branches are then summed. Because only a very small fraction of weight outliers are stored in FP16, under W4A4, the FP16 branch introduces only a 6% memory and a 7% runtime overhead.

**Conclusion.** The proposed outlier-aware quantizer effectively addresses the limitations posed by outlier activation and weight values. By employing random sampling for activation range estimation and selectively retaining a small fraction of weight outliers in higher precision, the method mitigates quantization errors without incurring substantial computational or storage costs. This dual strategy balances efficiency with accuracy, ensuring robust model performance under quantization.

### 3.3 FREQUENCY-AWARE CALIBRATION STRATEGY

Optimize quantization boundaries through training can improve the performance to some extent. However, the quantized model still exhibits performance degradation. In particular, quantization affects frequency components unevenly. For high-frequency structures such as edges and textures, the distortions are relatively minor due to their sharp RGB variations. However, in low- and mid-frequency regions with gradual intensity transitions, quantization severely reduces representational capacity, often manifesting as color banding. As illustrated in Fig. 5, the output of the quantized model exhibits pronounced banding artifacts in smooth areas.

To address the above problem, we propose a frequency-aware calibration strategy. This method optimizes the parameters of the quantizer by extracting the mid- and low-frequency components from both the model outputs and the ground truth, then optimizing the loss accordingly. Specifically, following previous work (Liu et al., 2025; Wang et al., 2024), we employ a dedicated convolution kernel $\boldsymbol{k}$ to extract the mid- and low-frequency components. The kernel $\boldsymbol{k}$ can be defined as:

$$\boldsymbol{k} = \begin{bmatrix} 1/16 & 1/8 & 1/16 \\ 1/8 & 1/4 & 1/8 \\ 1/16 & 1/8 & 1/16 \end{bmatrix}. \tag{10}$$

The frequency extraction process is an $L$-step iterative procedure based on kernel $\boldsymbol{k}$. Given a image $\mathbf{I}$, in the $i$-th step ($i = 1, 2, ..., L$), the extraction can be formalized as:

$$\mathbf{I}^i = E_{\boldsymbol{k}}^i(\mathbf{I}^{i-1}), \tag{11}$$

where $\mathbf{I}^0 = \mathbf{I}$, $\mathbf{I}^i$ is the outcome of the $i$-th step of extraction. $E_{\boldsymbol{k}}^i$ denotes an operation where the input is convolved with the kernel $\boldsymbol{k}$ and a dilation of $2^i$. The whole **frequency extraction** process can be written as a recursive function $F(\cdot)$, which is defined as follows:

$$F(\mathbf{I}, i) = \begin{cases} \mathbf{I}, & i = 0, \\ E_{\boldsymbol{k}}^i(F(\mathbf{I}, i-1)), & i \geq 1. \end{cases} \tag{12}$$

The outcome of the whole frequency extraction procedure is $F(\mathbf{I}, L)$. As shown in Fig. 2b, during the phase of optimizing quantization boundaries, we extract the low- and mid-frequency components from the output image and ground truth. Then following ESDNet (Yu et al., 2022), we directly reuse these losses: a pixel-wise $\mathcal{L}_1$ loss and a feature-based perceptual loss $\mathcal{L}_p$. The perceptual loss $\mathcal{L}_p$ is computed as the $L_1$ distance between the VGG16 feature (Yu et al., 2022).

Table 1: Ablation study on sampling-based activation quantizer.

(a) Quantizer bound calculation.

| Method | MinMax | Percentile | Sample |
|---|---|---|---|
| PSNR ↑ | 16.51 | 16.85 | **17.61** |
| SSIM ↑ | 0.5255 | 0.6701 | **0.6892** |
| LPIPS ↓ | 0.6786 | 0.4639 | **0.4552** |

(b) Smooth transformation.

| Method | Original | Smooth (Raw) | Smooth (Sample) |
|---|---|---|---|
| PSNR ↑ | 17.61 | 20.43 | **20.52** |
| SSIM ↑ | 0.6892 | 0.7408 | **0.7538** |
| LPIPS ↓ | 0.4552 | 0.3363 | **0.3247** |

Table 2: Ablation study on mixed-precision weight quantizer.

(a) Outlier handling method.

| Method | Baseline | Discard Outliers | Store Random | Store Outlier |
|---|---|---|---|---|
| PSNR ↑ | 20.52 | 16.29 | 20.54 | **20.92** |
| SSIM ↑ | 0.7538 | 0.6724 | 0.7542 | **0.7570** |
| LPIPS ↓ | 0.3247 | 0.4103 | 0.3244 | **0.3171** |

(b) Outliers Proportion ($\beta$).

| #$\beta$ | PSNR ↑ | SSIM ↑ | LPIPS ↓ | Params (M) | Ops (G) |
|---|---|---|---|---|---|
| 0.25% | 20.88 | 0.7565 | 0.3180 | 0.7837 | 1.78 |
| 0.50% | 20.92 | 0.7570 | 0.3171 | 0.7948 | 1.80 |
| 1.00% | 21.04 | 0.7578 | 0.3171 | 0.8171 | 1.85 |

Specifically, given the output of the quantitative model $\mathbf{I}'$ and the corresponding ground truth $\hat{\mathbf{I}}$, the extracted component of low and mid frequencies are defined as $\mathbf{I}'_l = F(\mathbf{I}', L)$ and $\hat{\mathbf{I}}_l = F(\hat{\mathbf{I}}, L)$, respectively. Then, we compute the loss $\mathcal{L}_{\text{total}}$ on the extracted frequencies as follows:

$$\mathcal{L}_{\text{total}} = \mathcal{L}_1(\mathbf{I}'_l, \hat{\mathbf{I}}_l) + \mathcal{L}_p(\mathbf{I}'_l, \hat{\mathbf{I}}_l). \tag{13}$$

We only optimize the quantization boundaries of activations while keeping all other parameters fixed. Besides, we find that when $L$ is small (*e.g.*, $L = 1$), more high-frequency information is preserved, whereas a larger $L$ (*e.g.*, $L = 5$) focuses only on low frequencies. Since our goal is to focus on both low- and mid-frequency regions, we set $L = 3$. Experiments in Sec. 4.2 (Tab. 3b) further validate this choice. Since the frequency extraction procedure module introduces only a few additional convolutions, the extra training-time overhead is negligible.

In summary, by incorporating mid- and low-frequency components into quantization boundary training, the proposed frequency-aware calibration strategy effectively alleviates the uneven impact of quantization across frequency bands. As shown in Fig. 5 (Mid&Low-Freq), this strategy significantly reduces banding artifacts caused by low-bit quantization.

## 4 EXPERIMENTS

### 4.1 EXPERIMENTAL SETTINGS

**Datasets and Metrics.** We conduct experiments on three moiré pattern removal datasets: UHDM (Yu et al., 2022), FHDMi (He et al., 2020), and LCDMoiré (Yuan et al., 2019). For each dataset, we randomly select 200 image pairs from the training dataset and crop them to construct a calibration dataset. After the calibration stage, we evaluate the results on the corresponding test dataset. For quantitative assessment, we adopt PSNR, SSIM (Wang et al., 2004), and LPIPS (Zhang et al., 2018).

**Implementation Details.** We adopt ESDNet (Yu et al., 2022) as the backbone and provide results under 3, 4, 6, and 8 bits. The quantization setup is denoted as W$w$A$a$, where $w$ and $a$ are the bit widths of weights and activations. For the sampling rate, we set $\gamma_1 = \gamma_2 = 10^{-3}$. For the smoothing factor, we use $\alpha = 0.5$. Both our method and the compared baselines apply static quantization, with per-channel quantization for weights and per-tensor quantization for activations. The baselines and our method are fairly evaluated, with more **details** provided in the supplementary materials.

**Calibration Settings.** During optimization, the quantization parameters are trained for 4 epochs using the Adam (Kingma & Ba, 2015) optimizer with $\beta_1 = 0.9$ and $\beta_2 = 0.999$. The image pairs are randomly cropped into $512 \times 512$, and the batch size is 1. The initial learning rate is set to 0.001 and scheduled with cyclic cosine annealing (Loshchilov & Hutter, 2017).

### 4.2 ABLATION STUDY

We investigate the impact of the proposed outlier-aware quantizer (for activations and weights) and the frequency-aware calibration strategy. All experiments are conducted on the UHDM (Yu et al., 2022) dataset with 4-bit quantization (W4A4). The experiment settings follow Sec. 4.1 to ensure fairness. Results are presented in Tabs. 1, 2, and 3, as well as Fig. 5.

Table 3: Ablation study on frequency-aware calibration strategy.

(a) Calibration strategy.

| Method | No Calibration | Orignal Image | High-Freq | Mid&Low-Freq |
|---|---|---|---|---|
| PSNR ↑ | 20.92 | 20.95 | 20.62 | **21.08** |
| SSIM ↑ | 0.7570 | 0.7604 | 0.7538 | **0.7626** |
| LPIPS ↓ | 0.3171 | 0.3088 | 0.3275 | **0.3068** |

(b) Frequency extraction level.

| Level | 1 | 3 | 5 |
|---|---|---|---|
| PSNR ↑ | 20.96 | **21.08** | 21.05 |
| SSIM ↑ | 0.7605 | **0.7626** | 0.7605 |
| LPIPS ↓ | 0.3092 | **0.3068** | 0.3071 |

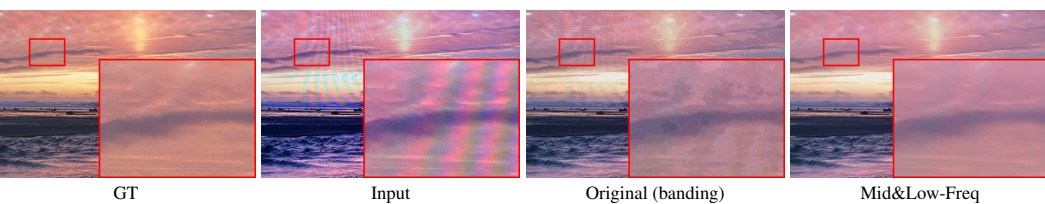

| GT | Input | Original (banding) | Mid&Low-Freq |
|---|---|---|---|

Figure 5: Visualization of results under different calibration strategies. Original: training in the spatial domain suffers from banding artifacts under low-bit quantization. Mid&Low-Freq: our frequency-aware calibration strategy leverages mid- and low-frequency to mitigate banding.

**Activation Quantizer.** We evaluate different strategies for calculating activation quantization bounds in Tab. 1a. The baseline is the min-max method (Jacob et al., 2018). We compare it with the percentile (Li et al., 2019) and our proposed sample-based method (as in Eq. 7). Compared with previous boundary selection, our sampling-based approach proves more effective.

Furthermore, we introduce a smooth transformation to adjust channels within the sample-based framework. As illustrated in Fig. 3, we compare two variants: the original definition and our sample-based smoothing. Results in Tab. 1b show that outlier removal via sampling enhances the effectiveness of the smoothing. In general, our sample-based activation quantizer leads to more robust quantization.

**Weight Quantizer.** We compare different strategies for handling weight outliers in Tab. 2a. We adopt the smooth (sample-based) result from Tab. 1b as the baseline. The results show that, compared with randomly storing some weights in FP16, explicitly preserving outliers in FP16 is more effective. Besides, consistent with the analysis in Sec. 3.2, discarding outliers leads to a performance drop.

We further evaluate the proportion of outliers ($\beta$) in Tab. 2b. The Ops are computed under an input size of $3 \times 224 \times 224$. We find that setting the ratio $\beta = 0.5\%$ achieves a favorable trade-off between efficiency and performance. Therefore, we adopt this setting as the default in QuantDemoire.

**Calibration Strategy.** We conduct an ablation study on different calibration strategies for optimizing quantized parameters. We apply the outlier-aware quantizer as the baseline. Results in Tab. 3a show that focusing on mid- and low-frequency components yields better performance.

We also evaluate different levels of frequency extraction (Tab. 3b). The relatively high (level=1) or low (level=5) frequencies are not optimal. In contrast, level=3 achieves better performance, which aligns with our analysis. Besides, we visualize results under different calibration strategies in Fig. 5. Our calibration strategy alleviates banding artifacts caused by quantization.

### 4.3 COMPARISON WITH STATE-OF-THE-ART METHODS

We utilize ESDNet (Yu et al., 2022) as the full-precision backbone and compare QuantDemoire with several advanced post-training quantization methods, including MinMax (Jacob et al., 2018), Percentile (Li et al., 2019), 2DQuant (Liu et al., 2024), and SVDQuant (Li et al., 2024).

**Quantitative Results.** Quantitative evaluations are summarized in Tab. 4. Across all datasets, bit-widths, and metrics, our approach consistently delivers superior performance. Notably, the improvements are more pronounced at lower bit-widths (*e.g.*, 4/3-bit). For example, under the 4-bit setting, our approach surpasses the latest method, SVDQuant (Li et al., 2024), by more than **6 dB** on the UHDM (Yu et al., 2022) dataset. Furthermore, our model retains performance close to the full-precision network under 8/6-bit settings, while the degradation at 3/4-bit is greatly reduced. More results and analyses (especially for SVDQuant) are provided in the supplementary material.

**Qualitative Results.** We present visual comparisons across 3-bit, 4-bit, and 6-bit settings on different datasets in Fig. 6. Existing quantization baselines tend to leave residual moiré artifacts or fail to reconstruct clean structures, especially on difficult samples and at lower precision. In contrast, our

Table 4: Quantitative comparison with state-of-the-art methods. The best and second-best results are colored red and blue. Our method outperforms on various datasets and metrics.

| Method | Bit | UHDM PSNR↑ | SSIM↑ | LPIPS↓ | FHDMi PSNR↑ | SSIM↑ | LPIPS↓ | LCDMoiré PSNR↑ | SSIM↑ | LPIPS↓ |
|---|---|---|---|---|---|---|---|---|---|---|
| ESDNet (Yu et al., 2022) | W32A32 | 22.12 | 0.7956 | 0.2551 | 24.50 | 0.8351 | 0.1354 | 44.83 | 0.9963 | 0.0097 |
| MinMax (Jacob et al., 2018) | W8A8 | 21.50 | 0.7727 | 0.2596 | 20.30 | 0.7631 | 0.2600 | 37.92 | 0.9865 | 0.0236 |
| Percentile (Li et al., 2019) | W8A8 | 19.38 | 0.7744 | 0.2784 | 19.73 | 0.7519 | 0.2734 | 30.11 | 0.9585 | 0.0281 |
| 2DQuant (Liu et al., 2024) | W8A8 | 21.20 | 0.7827 | 0.2749 | 20.57 | 0.7861 | 0.2034 | 41.60 | 0.9923 | 0.0214 |
| SVDQuant (Li et al., 2024) | W8A8 | 21.80 | 0.7907 | 0.2580 | 22.07 | 0.7966 | 0.1598 | 41.18 | 0.9915 | 0.0165 |
| QuantDemoire (ours) | W8A8 | 22.00 | 0.7932 | 0.2555 | 22.23 | 0.8026 | 0.1591 | 42.16 | 0.9930 | 0.0126 |
| MinMax (Jacob et al., 2018) | W6A6 | 20.48 | 0.7648 | 0.2828 | 19.85 | 0.7362 | 0.3040 | 24.53 | 0.9337 | 0.1692 |
| Percentile (Li et al., 2019) | W6A6 | 18.44 | 0.7562 | 0.2957 | 19.22 | 0.7353 | 0.2953 | 22.46 | 0.9342 | 0.1074 |
| 2DQuant (Liu et al., 2024) | W6A6 | 20.02 | 0.7595 | 0.2893 | 20.66 | 0.7389 | 0.2558 | 26.53 | 0.9366 | 0.2241 |
| SVDQuant (Li et al., 2024) | W6A6 | 20.86 | 0.7602 | 0.3015 | 20.91 | 0.6883 | 0.3732 | 32.10 | 0.9691 | 0.1314 |
| QuantDemoire (ours) | W6A6 | 21.61 | 0.7874 | 0.2572 | 21.63 | 0.7861 | 0.1721 | 40.48 | 0.9910 | 0.0206 |
| MinMax (Jacob et al., 2018) | W4A4 | 16.51 | 0.5255 | 0.6786 | 16.01 | 0.4958 | 0.6686 | 15.14 | 0.7031 | 0.7500 |
| Percentile (Li et al., 2019) | W4A4 | 16.85 | 0.6701 | 0.4639 | 16.92 | 0.6424 | 0.4858 | 14.95 | 0.8290 | 0.5596 |
| 2DQuant (Liu et al., 2024) | W4A4 | 17.07 | 0.6288 | 0.5117 | 14.80 | 0.3579 | 0.7107 | 15.11 | 0.6075 | 0.7561 |
| SVDQuant (Li et al., 2024) | W4A4 | 14.68 | 0.5762 | 0.6559 | 12.43 | 0.2818 | 0.7961 | 15.25 | 0.7201 | 0.6687 |
| QuantDemoire (ours) | W4A4 | 21.08 | 0.7626 | 0.3068 | 20.28 | 0.6745 | 0.3369 | 31.28 | 0.9544 | 0.1793 |
| MinMax (Jacob et al., 2018) | W3A3 | 12.53 | 0.3802 | 0.8630 | 12.16 | 0.2903 | 0.8514 | 10.11 | 0.6051 | 0.8376 |
| Percentile (Li et al., 2019) | W3A3 | 14.79 | 0.5206 | 0.6869 | 14.33 | 0.4444 | 0.7496 | 9.30 | 0.5242 | 1.0272 |
| 2DQuant (Liu et al., 2024) | W3A3 | 11.20 | 0.2465 | 0.8460 | 10.92 | 0.2879 | 0.8612 | 9.07 | 0.3026 | 1.0153 |
| SVDQuant (Li et al., 2024) | W3A3 | 14.83 | 0.4547 | 0.7289 | 9.34 | 0.2127 | 0.8840 | 10.17 | 0.5658 | 0.9753 |
| QuantDemoire (ours) | W3A3 | 19.12 | 0.6839 | 0.4567 | 18.26 | 0.5722 | 0.4930 | 22.02 | 0.8558 | 0.4544 |

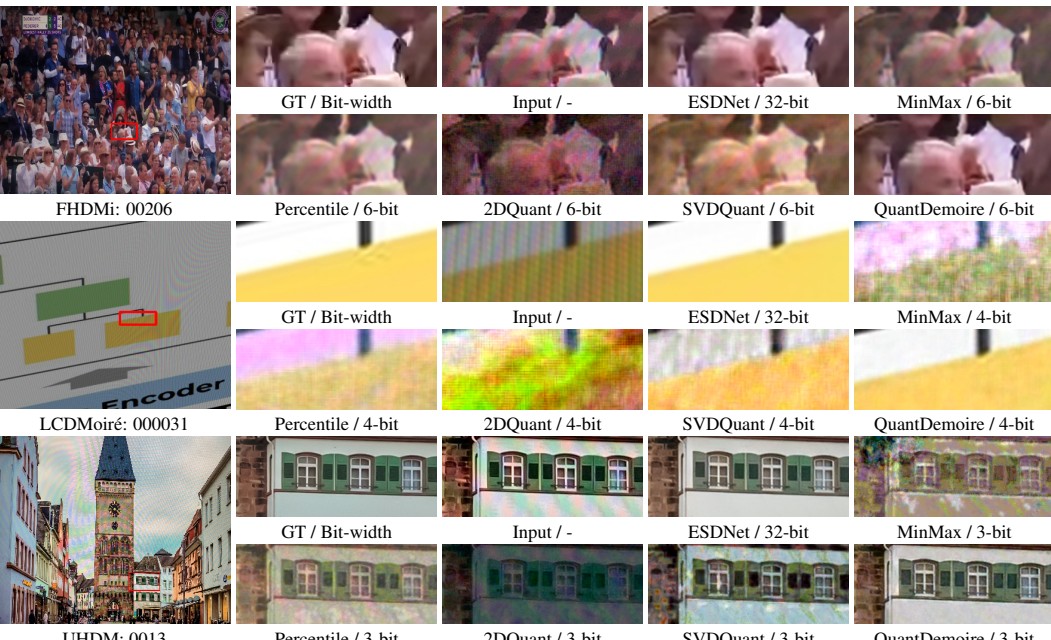

Figure 6: Visual comparison on UHDM (Yu et al., 2022), FHDMi (He et al., 2020), and LCD-Moiré (Yuan et al., 2019) datasets. Our QuantDemoire outperforms other quantization methods.

QuantDemoire effectively removes moiré patterns and restores fine image details. Meanwhile, our QuantDemoire maintains results that are very close to full precision, even at low bit-widths (*e.g.*, 3-bit). This observation is consistent with the quantitative results in Tab. 4. More visual results on various datasets and bit settings are provided in the supplementary material.

**Compression Ratio.** We report the parameter (Params) and operation (Ops) compression ratios at 6/4/3-bit in Tab. 5. The Ops are measured with an input size of $3 \times 224 \times 224$. Compared with the full-precision (32-bit) model, our approach significantly reduces both model size and computational cost. At 4-bit, the compression rate exceeds 86.6%, while the performance drop is limited to 4.7% (PSNR). Moreover, compared with existing quantization methods, QuantDemoire demonstrates clear advantages. For instance, compared to 2DQuant, it achieves over a 4 dB gain with only a negligible increase in overhead. More results on compression ratios are provided in the supplementary material.

Table 5: Compression ratios of Params and Ops. Ops are measured with an input size of $3 \times 224 \times 224$.

| Method | Bit (w/a) | Params (M) (↓Ratio) | Ops (G) (↓Ratio) | UHDM PSNR ↑ | SSIM ↑ |
|---|---|---|---|---|---|
| ESDNet | 32/32 | 5.93 (↓0%) | 13.52 (↓0%) | 22.12 | 0.7956 |
| 2DQuant | 6/6 | 1.12 (↓81.14%) | 2.54 (↓81.25%) | 20.02 | 0.7595 |
| SVDQuant | 6/6 | 1.45 (↓75.53%) | 4.60 (↓65.97%) | 20.86 | 0.7602 |
| QuantDemoire | 6/6 | 1.16 (↓80.43%) | 2.64 (↓80.49%) | 21.61 | 0.7874 |
| 2DQuant | 4/4 | 0.75 (↓87.38%) | 1.69 (↓87.50%) | 17.07 | 0.6288 |
| SVDQuant | 4/4 | 1.08 (↓81.78%) | 3.76 (↓72.19%) | 14.68 | 0.5762 |
| QuantDemoire | 4/4 | 0.79 (↓86.61%) | 1.80 (↓86.68%) | 21.08 | 0.7626 |
| 2DQuant | 3/3 | 0.56 (↓90.51%) | 1.27 (↓90.63%) | 11.20 | 0.2465 |
| SVDQuant | 3/3 | 0.90 (↓84.89%) | 3.34 (↓75.30%) | 14.83 | 0.4547 |
| QuantDemoire | 3/3 | 0.61 (↓89.69%) | 1.38 (↓89.78%) | 19.12 | 0.6839 |

## 5 CONCLUSION

In this paper, we propose QuantDemoire, a post-training quantization method tailored for image demoiréing. Our design is developed from two perspectives: the quantizer and the calibration strategy. We first introduce the outlier-aware quantizer to reduce quantization errors caused by outliers. We Besides, we develop a frequency-aware calibration strategy emphasizing mid- and low-frequency components to mitigate banding artifacts caused by low-bit quantization. Comprehensive experiments verify that our method effectively reduces overhead compared with the full-precision model. Meanwhile, QuantDemoire outperforms current advanced quantization methods.

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
