# QuantDemoire: Quantization with Outlier Aware for Image Demoiréing
## *Supplementary Material*

## 1 More Ablation

Table 1: Ablation on the sampling rate in the activation. Evaluation is conducted on UDHM, 4-bit.



(a) Smooth sampling rate ($\gamma_1$).

| Rate | $10^{-2}$ | $10^{-3}$ | $10^{-4}$ |
|------|-----------|-----------|-----------|
| PSNR ↑ | 19.92 | **20.92** | 20.80 |
| SSIM ↑ | 0.6941 | **0.7570** | 0.7505 |
| LPIPS ↓ | 0.4141 | **0.3171** | 0.3221 |

(b) Bound sampling rate ($\gamma_2$).

| Rate | $10^{-2}$ | $10^{-3}$ | $10^{-4}$ |
|------|-----------|-----------|-----------|
| PSNR ↑ | 20.88 | **20.92** | 20.89 |
| SSIM ↑ | 0.7558 | **0.7570** | 0.7474 |
| LPIPS ↓ | 0.3183 | **0.3171** | 0.3388 |



For the smooth stage ($\gamma_1$), we ablate the sampling rate by varying $\gamma_1$ while fixing $\gamma_2 = 10^{-3}$. As shown in Tab. 1a, the setting that produces the best performance corresponds to $\gamma_1 = 10^{-3}$. Higher rates introduce excessive outliers in the smoothing process, whereas lower rates bias the preliminary estimate. A moderate sampling rate, therefore, provides the most stable and accurate range estimation.

For the bound stage ($\gamma_2$), we perform an analogous ablation by varying $\gamma_2$ while fixing $\gamma_1 = 10^{-3}$. Tab. 1b indicates that $\gamma_2 = 10^{-3}$ gives the best results. Larger rates lead to the accumulation of outliers during refinement, and smaller rates cause biased bounds. These results confirm that a moderate sampling rate also optimizes accuracy, reinforcing our overall design choice.

## 2 Implementation Details of Comparison Methods

**MinMax.** We apply per-channel quantization to the weights and per-tensor quantization to the activations. During the calibration stage, we directly use the minimum and maximum activation values of the activation tensor as the quantization boundaries.

**Percentile.** We also use per-channel quantization for the weights and per-tensor quantization for the activations. Specifically, for both weights and activations, the 99.9%-th and 0.1%-th percentiles of all values are consistently used as the quantization boundaries.

**2DQuant.** We first employ the DOBI (Liu et al., 2024) method to search for quantizer parameters. Then we train the model using the quantization parameters obtained from DQC (Liu et al., 2024), with a training configuration of 100 epochs, a batch size of 4, and a learning rate of $10^{-2}$. All other training settings remain consistent with those used in our proposed method.

**SVDQuant.** We first apply SmoothQuant (Xiao et al., 2023) to mitigate outliers in the activations. Next, we perform an SVD decomposition on the weights and compute both the low-rank and residual branches. To ensure fairness in our experiments and to prevent the low-rank branch from introducing an excessive number of additional parameters, we set the rank to 2.

## 3 Variants of SVDQuant

We observed that SVDQuant suffers from a substantial performance drop when quantized to extremely low bit-widths (3 and 4 bits). To investigate this issue, we conducted further experiments with higher ranks. The results in Tab. 2 show that when the rank is increased to 8, SVDQuant achieves a significant performance improvement. However, when the rank is set to 8, although the performance improves, the relatively small parameter size of the convolutional layers in ESDNet leads SVDQuant to introduce an excessive number of additional parameters. Therefore, in our experiments, we set the rank of SVDQuant to 2 for comparative fairness (parameter).

Table 2: Results of SVDQuant (rank = 2 and 8). Evaluation is conducted on UDHM, 3, and 4-bit.

| Method | Bit($w/a$) | Params (M) | Ops (G) | UHDM | | |
|---|---|---|---|---|---|---|
| | | | | PSNR ↑ | SSIM ↑ | LPIPS ↓ |
| SVDQuant (rank = 2) | 4/4 | 1.08 | 3.76 | 14.68 | 0.5762 | 0.6559 |
| SVDQuant (rank = 8) | 4/4 | 2.10 | 8.48 | 18.69 | 0.7428 | 0.3615 |
| QuantDemoire (ours) | 4/4 | **0.79** | **1.80** | **21.08** | **0.7626** | **0.3068** |
| SVDQuant (rank = 2) | 3/3 | 0.90 | 3.34 | 14.83 | 0.4547 | 0.7289 |
| SVDQuant (rank = 8) | 3/3 | 2.01 | 6.80 | 17.20 | 0.6707 | 0.5033 |
| QuantDemoire (ours) | 3/3 | **0.61** | **1.38** | **19.12** | **0.6839** | **0.4567** |

Table 3: Compression ratios of Params and Ops at 8/6/4/3-bit. Ops are measured with an input size of $3 \times 224 \times 224$. Our QuantDemoire maintains efficiency and performance.

| Method | Bit ($w/a$) | Params (M) (↓Ratio) | Ops (G) (↓Ratio) | UHDM | | |
|---|---|---|---|---|---|---|
| | | | | PSNR ↑ | SSIM ↑ | LPIPS ↓ |
| ESDNet (Yu et al., 2022) | 32/32 | 5.93 (↓0%) | 13.52 (↓0%) | 22.12 | 0.7956 | 0.2551 |
| MinMax (Jacob et al., 2018) | 8/8 | 1.49 (↓74.90%) | 3.38 (↓75.00%) | 21.50 | 0.7727 | 0.2596 |
| Percentile (Li et al., 2019) | 8/8 | 1.49 (↓74.90%) | 3.38 (↓75.00%) | 19.38 | 0.7744 | 0.2784 |
| 2DQuant (Liu et al., 2024) | 8/8 | 1.49 (↓74.90%) | 3.38 (↓75.00%) | 21.20 | 0.7827 | 0.2749 |
| SVDQuant (Li et al., 2024) | 8/8 | 1.82 (↓69.29%) | 5.44 (↓59.75%) | 21.80 | 0.7907 | 0.2580 |
| QuantDemoire (ours) | 8/8 | 1.53 (↓74.25%) | 3.47 (↓74.31%) | 22.00 | 0.7932 | 0.2555 |
| MinMax (Jacob et al., 2018) | 6/6 | 1.12 (↓81.14%) | 2.54 (↓81.25%) | 20.48 | 0.7648 | 0.2828 |
| Percentile (Li et al., 2019) | 6/6 | 1.12 (↓81.14%) | 2.54 (↓81.25%) | 18.44 | 0.7562 | 0.2957 |
| 2DQuant (Liu et al., 2024) | 6/6 | 1.12 (↓81.14%) | 2.54 (↓81.25%) | 20.02 | 0.7595 | 0.2893 |
| SVDQuant (Li et al., 2024) | 6/6 | 1.45 (↓75.53%) | 4.60 (↓65.97%) | 20.86 | 0.7602 | 0.3015 |
| QuantDemoire (ours) | 6/6 | 1.16 (↓80.43%) | 2.64 (↓80.49%) | 21.61 | 0.7874 | 0.2572 |
| MinMax (Jacob et al., 2018) | 4/4 | 0.75 (↓87.38%) | 1.69 (↓87.50%) | 16.51 | 0.5255 | 0.6786 |
| Percentile (Li et al., 2019) | 4/4 | 0.75 (↓87.38%) | 1.69 (↓87.50%) | 16.85 | 0.6701 | 0.4639 |
| 2DQuant (Liu et al., 2024) | 4/4 | 0.75 (↓87.38%) | 1.69 (↓87.50%) | 17.07 | 0.6288 | 0.5117 |
| SVDQuant (Li et al., 2024) | 4/4 | 1.08 (↓81.78%) | 3.76 (↓72.19%) | 14.68 | 0.5762 | 0.6559 |
| QuantDemoire (ours) | 4/4 | 0.79 (↓86.61%) | 1.80 (↓86.68%) | 21.08 | 0.7626 | 0.3068 |
| MinMax (Jacob et al., 2018) | 3/3 | 0.56 (↓90.51%) | 1.27 (↓90.63%) | 12.53 | 0.3802 | 0.8630 |
| Percentile (Li et al., 2019) | 3/3 | 0.56 (↓90.51%) | 1.27 (↓90.63%) | 14.79 | 0.5206 | 0.6869 |
| 2DQuant (Liu et al., 2024) | 3/3 | 0.56 (↓90.51%) | 1.27 (↓90.63%) | 11.20 | 0.2465 | 0.8460 |
| SVDQuant (Li et al., 2024) | 3/3 | 0.90 (↓84.89%) | 3.34 (↓75.30%) | 14.83 | 0.4547 | 0.7289 |
| QuantDemoire (ours) | 3/3 | 0.61 (↓89.69%) | 1.38 (↓89.78%) | 19.12 | 0.6839 | 0.4567 |

## 4 MORE COMPRESSION RATIO

A more comprehensive comparison of compression ratios is presented in Tab. 3. Our method achieves performance close to that of the full-precision model, while maintaining a high compression ratio.

## 5 MORE QUALITATIVE RESULTS

Additional visual comparisons are presented in Figs. 2 and 3. Our method demonstrates clear advantages under the 3-, 4-, and 6-bit settings, as well as across all datasets.

## 6 MORE DISTRIBUTION VISUALIZATIONS

Additional distributions of weights and activations are presented in Figs. 4, 5, 6, and 7. It can be observed that the activations in most convolutional layers approximately follow either an exponential or a Gaussian distribution, while the weights in the majority of convolutional layers exhibit an approximately Gaussian distribution overall. This further proves our point.

## 7 PSEUDOCODE OF QUANTDEMOIRE

To provide a clearer description of the procedure of our method, we present the following pseudocode in Alg. 1 for the sampling-based activation quantizer.

---

**Algorithm 1:** Pipeline of Sampling-Based Activation Quantizer

---

**Initialization:**
$model \leftarrow \text{ESDNet(fp32)}$
$N \leftarrow$ number of batches in calibration dataset
**for** *convolution layers $L_i$ in model* **do**
   $W_i \leftarrow$ pretrained weight of $L_i$
   $C_i \leftarrow$ number of input channels of $L_i$
   $lb_a^i \leftarrow 0, \; ub_a^i \leftarrow 0$
   $lb_w^i \leftarrow 0, \; ub_w^i \leftarrow 0$
   $\boldsymbol{s}^i[1:C_i] \leftarrow 0$
**end**
**Calibration Stage 1:**
**for** *x in calibration dataset* **do**
   $model(x)$
   **for** *convolution layers $L_i$ in model* **do**
      $\mathbf{A} \leftarrow$ latest activation tensor of $L_i$
      $\boldsymbol{s}^i[c] \leftarrow \max\left|\text{Sample}_\gamma(\mathbf{A}_c)\right|, \quad \forall c = 1, \ldots, C_i$
   **end**
**end**
**for** *convolution layers $L_i$ in model* **do**
   $\boldsymbol{s}^i \leftarrow \boldsymbol{s}^i/N$
**end**
**Calibration Stage 2:**
**for** *x in calibration dataset* **do**
   $model(x)$
   **for** *convolution layers $L_i$ in model* **do**
      $\mathbf{A} \leftarrow$ latest activation tensor of $L_i$
      $lb_a^i \leftarrow lb_a^i + \min(\text{Sample}_\gamma(\mathbf{A} \oslash \boldsymbol{s}^i))$
      $ub_a^i \leftarrow ub_a^i + \max(\text{Sample}_\gamma(\mathbf{A} \oslash \boldsymbol{s}^i))$
   **end**
**end**
**for** *convolution layers $L_i$ in model* **do**
   $lb_a^i \leftarrow lb_a^i/N$
   $ub_a^i \leftarrow ub_a^i/N \; W_i \leftarrow W_i \oslash \boldsymbol{s}^i$
**end**

---

# 8 REBUTTAL DETAILS

## 8.1 THEORETICAL ANALYSIS OF SAMPLING APPROACH

Based on the following analysis, we will demonstrate that: (1) under the activation distribution of the demoiréing model, the sampling-based estimation is equivalent to percentile clipping with a dynamically varying percentile that adapts to the number of activations, and this dynamic adjustment better aligns with the actual distribution characteristics of outliers; (2) the variance of the equivalent percentile is small, thereby demonstrating the stability of our method.

### 8.1.1 EQUIVALENCE BETWEEN SAMPLING AND DYNAMIC PERCENTILE CLIPPING

As illustrated in Fig. 4 and Fig. 5 of the supplementary material, the activation distributions of nearly all convolutional layers approximate a normal distribution following the smoothing operation. Consequently, we model elements $N$ within a given activation tensor as $N$ independent and identically distributed random variables (i.i.d.) drawn from a standard normal distribution. We used a sampling ratio of $\gamma = 0.1\%$, resulting in a sample size of $n = \gamma N$. Formally, let $X_1, \ldots, X_N \overset{\text{i.i.d.}}{\sim} \mathcal{N}(0, 1)$ and denote the maximum of all values by $M_N = \max_{1 \leq i \leq N} X_i$. Its probability density function (PDF):

Table 4: Results of number of extreme outliers $|x - mean| > 4 * std$ means the number of activations that lies more than **four** standard deviations away from the mean, and $|x - mean| > 4.5 * std$ has similar meaning.

| N | $|x - mean| > 4 * std$ | $|x - mean| > 4.5 * std$ |
|---|---|---|
| $10^6$ | 51 | 17 |
| $10^7$ | 103 | 41 |
| $10^8$ | 253 | 72 |

$$f_{M_N}(x) = N\phi(x)\Phi(x)^{N-1},\tag{1}$$

where $\phi$ and $\Phi$ are the standard normal PDF and cumulative distribution function (CDF), respectively. The expectation is then:

$$\mathbb{E}[M_N] = \int_{-\infty}^{\infty} xN\phi(x)\Phi(x)^{N-1}dx.\tag{2}$$

Similarly, the expected maximum of the randomly selected activation values $n$ is:

$$\mathbb{E}[M_n] = \int_{-\infty}^{\infty} xn\phi(x)\Phi(x)^{n-1}dx.\tag{3}$$

This allows us to treat the sampling-based estimation as equivalent to clipping at a percentile threshold, $p$, for our sampling-based estimation using the CDF: $p = \Phi(\mathbb{E}[M_n])$. We observe that as the total number of elements ($N$) increases, the sampling-based estimation corresponds to a higher equivalent percentile threshold. For example, for $N = 10^6$, $p \approx 99.95\%$; for $N = 10^7$, $p \approx 99.995\%$; and for $N = 10^8$, $p \approx 99.9995\%$. Furthermore, we find that due to the smoothing operation, the number of extreme outliers tends to stabilize for very large $N$ ($N > 10^6$) rather than growing linearly, as shown in Tab 4:

Therefore, by dynamically clipping at a higher percentile for larger $N$, our sampling-based estimation effectively removes extreme outliers while avoiding the pitfall of fixed-percentile clipping, which may incorrectly truncate non-outlier values.

### 8.1.2 VARIANCE ANALYSIS

Based on this analysis, we proceed to calculate the standard deviation of $p$. Using a first-order Taylor expansion (the Delta Method), the standard deviation can be approximated as:

$$\text{std}(p) = \text{std}(\Phi(M_n)) \approx [\phi(\mathbb{E}[M_n])]\text{std}(M_n),$$
$$\mathbb{E}[M_n^2] = \int_{-\infty}^{\infty} x^2 f_{M_n}(x)dx = \int_{-\infty}^{\infty} x^2 n\phi(x)\Phi(x)^{n-1}dx,\tag{4}$$

Numerical calculations for $N = 10^6, 10^7$ and $10^8$ yield standard deviations of $6.522 \times 10^{-4}$, $6.596 \times 10^{-5}$, and $6.653 \times 10^{-6}$, respectively, which are negligible.

### 8.2 ADDITIONAL EXPERIMENTAL RESULTS

### 8.2.1 STABILITY AND REPRODUCIBILITY

To validate the stability of our method, we assessed the variance by conducting five runs as shown in Tab. 5.

The standard deviation of each metric across five runs is negligible, demonstrating the stability of our method.

Table 5: Results of PSNR, SSIM, and LPIPS metrics across 5 runs

| RUN | PSNR ↑ | SSIM ↑ | LPIPS ↓ |
|---|---|---|---|
| 1 | 21.0808 | 0.7626 | 0.3068 |
| 2 | 21.0835 | 0.7623 | 0.3066 |
| 3 | 21.0798 | 0.7628 | 0.3070 |
| 4 | 21.0801 | 0.7624 | 0.3059 |
| 5 | 21.0824 | 0.7631 | 0.3062 |
| std | 0.0014 | 0.0003 | 0.0004 |

### 8.2.2 MORE COMPARISON WITH FIXED PERCENTILE

We present additional experiments demonstrating that our sampling method outperforms fixed-percentile clipping. Specifically, we compare sampling with 95th- and 99.9th-percentile clipping. The results in Tab. 6 show that sampling provides clear advantages over fixed-percentile clipping.

Table 6: Comparison of different methods on PSNR, SSIM, and LPIPS metrics. Apart from using sampling-based estimation, % clipping, and 99.9% clipping respectively to compute the quantization boundaries and the smoothing factor, the remaining settings for all three methods are identical to the final configuration of our method.

| Method | PSNR | SSIM | LPIPS |
|---|---|---|---|
| Ours (sampling) | 20.2803 | 0.6745 | 0.3369 |
| percentile 95% | 19.3058 | 0.6026 | 0.3623 |
| percentile 99.9% | 19.3526 | 0.6091 | 0.3487 |

### 8.2.3 INITIALIZATION VS. OPTIMIZATION

Because rounding operations in quantization can destabilize training, training alone can only reduce quantization error within a small range; a well-chosen initialization of the quantization boundary remains crucial. The sampled-activation quantizer approach aims to initialize the quantization boundary to a favorable starting point, whereas boundary optimization subsequently performs fine-grained refinement.

Table 7: Comparison of different initialization of the quantization boundary

| Method | PSNR | SSIM | LPIPS |
|---|---|---|---|
| Ours (sampling) | 20.2803 | 0.6745 | 0.3369 |
| percentile 95% | 19.4556 | 0.6223 | 0.3601 |
| percentile 99.9% | 19.5781 | 0.6391 | 0.3532 |
| MinMax | 18.9265 | 0.5921 | 0.4013 |

The experiments in Tab. 7 substantiate this claim. We initialize the quantization boundaries using different methods, keep all other settings the same, and train each method for a sufficient number of epochs. The results show that initializing the quantization boundaries with sampling-based estimation offers a clear advantage.

This two-stage paradigm—good initialization followed by fine-grained refinement—is already established in the literature, e.g., in 2DQuant (Liu et al., 2024).

### 8.3 BANDING ARTIFACTS VISUALIZATION

Traditional image quality assessment (IQA) metrics do not accurately reflect the impact of banding artifacts, as illustrated in (Wang et al., 2016). Therefore, the advantage of frequency-aware calibration in IQA metrics is not significant. To visually demonstrate the effectiveness of frequency-aware

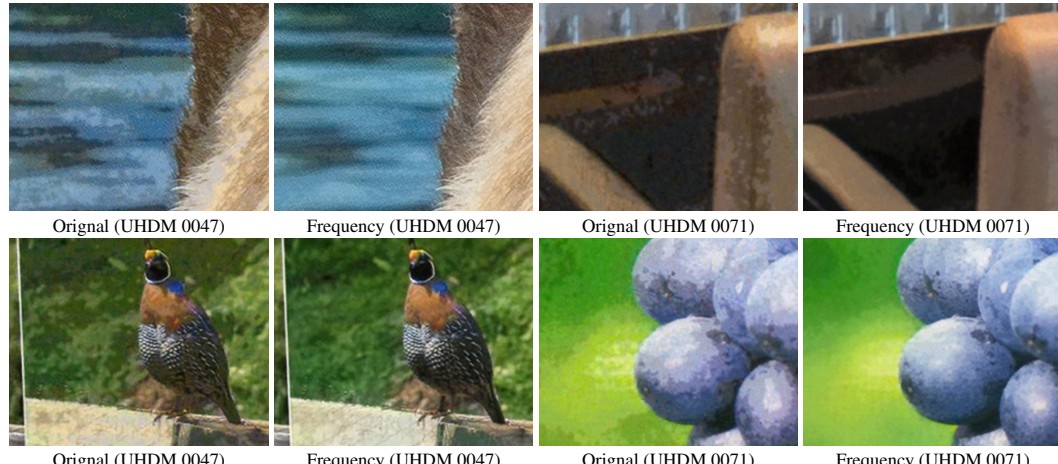

Figure 1: Visualization of the effectiveness of frequency-aware calibration. "Orignal" are the results of model trained with original loss function, "Frequency" are the results of model trained with frequency-aware calibration, which significantly weakened banding artifacts.

calibration, we present a comparison of the visual results from the model trained with original loss and frequency-aware calibration.

It is evident that under the extremely low bit-width setting of 4-bit, the baseline model (using the original loss) still suffers from severe banding artifacts. In contrast, frequency-aware calibration significantly alleviates this issue.

## 8.4 IMPLEMENTATION DETAILS OF THE FP16 BRANCH

Our Mixed-Precision Weight Quantizer retains only 1% of convolutional weights, creating extreme unstructured sparsity that is inherently unfriendly to parallel execution. We address this challenge with a specialized GPU kernel. We employ CSR-style per-channel indexing for sparse weight storage. The computation strategy tiles output rows across CUDA blocks and stripes threads along the column dimension, with each kernel serially traversing the sparse weights and accumulating contributions to the output tensor.

This design choice is justified by the extreme sparsity regime: for a typical 32×16×3×3 kernel, each output channel retains only 1-2 weight elements on average, making serial traversal more efficient than parallel scheduling overhead. This approach successfully exploits extreme unstructured sparsity while incurring only 6% GPU memory and 7% runtime overhead.

## 8.5 COMPREHENSIVE ABLATION STUDY DETAILS

Here we present the full experimental settings for all ablation studies reported in the main text:

All ablation experiments were conducted using static quantization.

**Table 1 (a) Quantizer bound calculation:** All methods are evaluated without smoothing and without any training-based optimization. Weights are quantized using MinMax. For activation quantization: MinMax uses the absolute minimum and maximum; Percentile applies 99.9th-percentile clipping; Sample uses the sampling-based estimator proposed in this paper.

**Table 1 (b) Smooth transformation:** In this setting, weights are quantized by MinMax, activations are quantized using the sampling-based estimator, and no training-based optimization is applied. Original applies no smoothing; Smooth (Raw) computes the smoothing factor from the absolute maximum activation; Smooth (Sample) computes the smoothing factor using the sampling-based estimator.

**Table 2 (a) Outlier handling method:** Baseline and Smooth (Sample) in Table 1 (b) share identical settings. Discard Outliers directly truncates weight outliers; Store Random randomly stores 1% of weights in FP16; Store Outlier stores the largest 0.5% and smallest 0.5% of weight elements.

**Table 2 (b) Outliers Proportion:** 0.50% is identical to Store Outlier in Table 2 (a); 0.25% stores the largest 0.25% and smallest 0.25% of weight elements (all other settings identical to the 0.50% case); 1% stores the largest 1% and smallest 1% of weight elements (all other settings identical to the 0.50% case).

**Table 3 (a) Calibration strategy:** No Calibration uses the same settings as Store Outlier in Table 2 (a). Original Image applies $L_1$ & $L_p$ calibration directly on the original images (on top of No Calibration). High-Freq performs calibration using high-frequency components of the images. Mid&Low-Freq performs calibration using mid- and low-frequency components.

**Table 3 (b) Frequency extraction level:** For level = 3, the settings are identical to Mid&Low-Freq in Table 3 (a). For level = 1 and level = 5, we use 1 and 5 convolution kernels, respectively, to extract frequency components; all other settings are the same as for level = 3

## 8.6 Comparison with Related Methods

### 8.6.1 Comparison with OmniQuant

Both the outlier-aware quantizer and OmniQuant (Shao et al., 2023) follow the same high-level smoothing-and-clipping scheme. However, OmniQuant (Shao et al., 2023) is designed to quantize large language models (LLMs), whereas the outlier-aware quantizer targets an image demoiréing model composed mainly of convolutional modules. Consequently, the two approaches differ substantially in their specific design choices.

**(1) Different clipping targets.** OmniQuant (Shao et al., 2023) employs learnable weight clipping in LWC. By contrast, in convolutional modules the weights are confined to small kernels and are relatively few per layer compared with the volume of activations. Directly truncating weight outliers can therefore cause disproportionate accuracy loss. Therefore, in the outlier-aware quantizer we clip only the activation quantization bounds rather than clipping the weights.

**(2) Choice of learnable parameters.** OmniQuant (Shao et al., 2023) treats both the smoothing factor and the weight clipping bounds as learnable. In our outlier-aware quantization, only the activation clipping bound is learnable; the smoothing factor is initialized using a sampling-based estimator and then kept fixed. Because the rounding operation in quantization can destabilize training, we observe that our design outperforms the alternative that makes both the smoothing factor and the weight clipping bounds learnable, as the results shown in Tab. 8.

Table 8: Comparison of only learning the clipping bound and learn clipping bound & smoothing factor. Experiments are conducted in 4bit, UHDM dataset.

| Method | PSNR | SSIM | LPIPS | Training epoch |
|---|---|---|---|---|
| Ours (only learn the clipping bound) | 21.0808 | 0.7626 | 0.3068 | 4 |
| Omniquant (learn clipping bound and smoothing factor) | 21.0643 | 0.7613 | 0.3106 | 40 |

Furthermore, by initializing the smoothing factor to a good value via sampling and reducing the number of learnable parameters, our method requires only 4 training epochs, whereas the configuration with learnable smoothing factor and clipping bounds typically requires 40 epochs. Consequently, our approach substantially reduces the training time.

### 8.6.2 Comparison with other mixed-precision methods

Indeed, many prior methods adopt mixed-precision quantization. For example, LLM.int8 (Dettmers et al., 2022) processes a very small number of outlier feature dimensions in FP16, and SqueezeLLM (Kim et al., 2023) stores a small subset of weights in a sparse format. However, few approaches are designed specifically for convolutions. Because outlier weights in convolution kernels occur sporadically, existing coarse-grained mixed-precision schemes are not well suited, and directly converting convolutions into sparse matrix multiplications introduces additional overhead. To address this, we implement a dedicated kernel for the FP16 branch that parallelizes processing of FP16 weights at the element level.

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

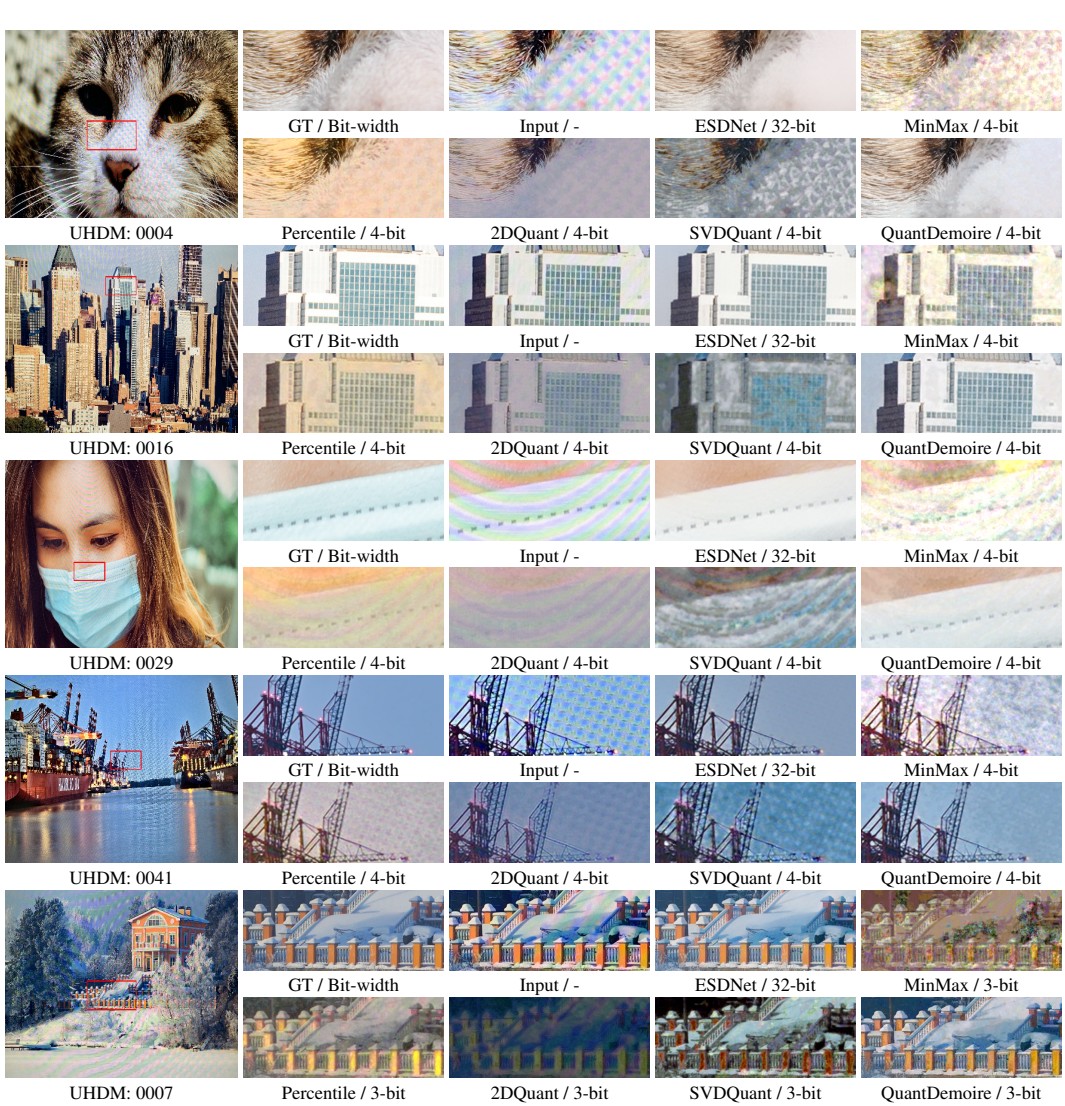

Figure 2: Visual comparison on the UHDM (Yu et al., 2022) dataset.

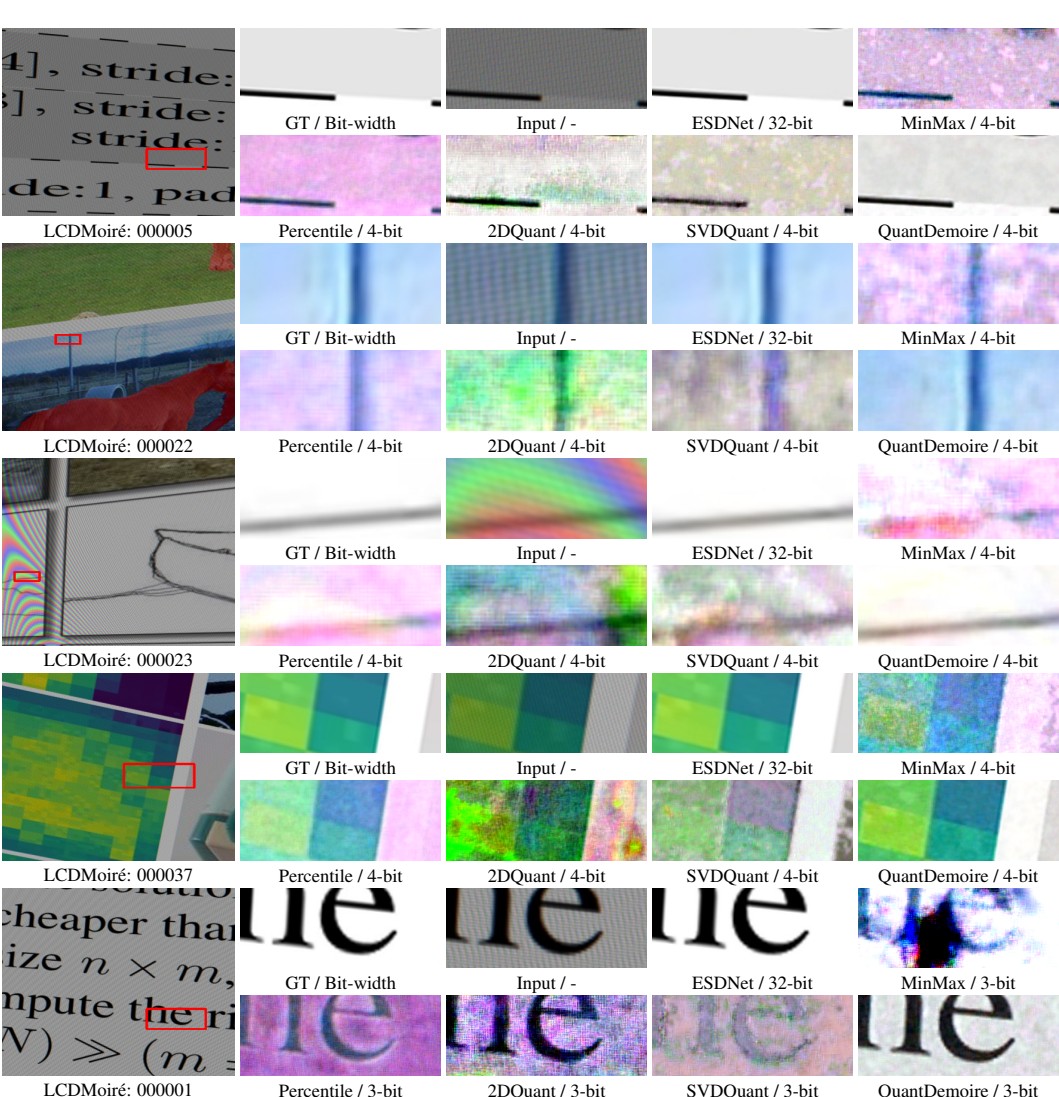

Figure 3: Visual comparison on the LCDMoiré (Yuan et al., 2019) dataset.

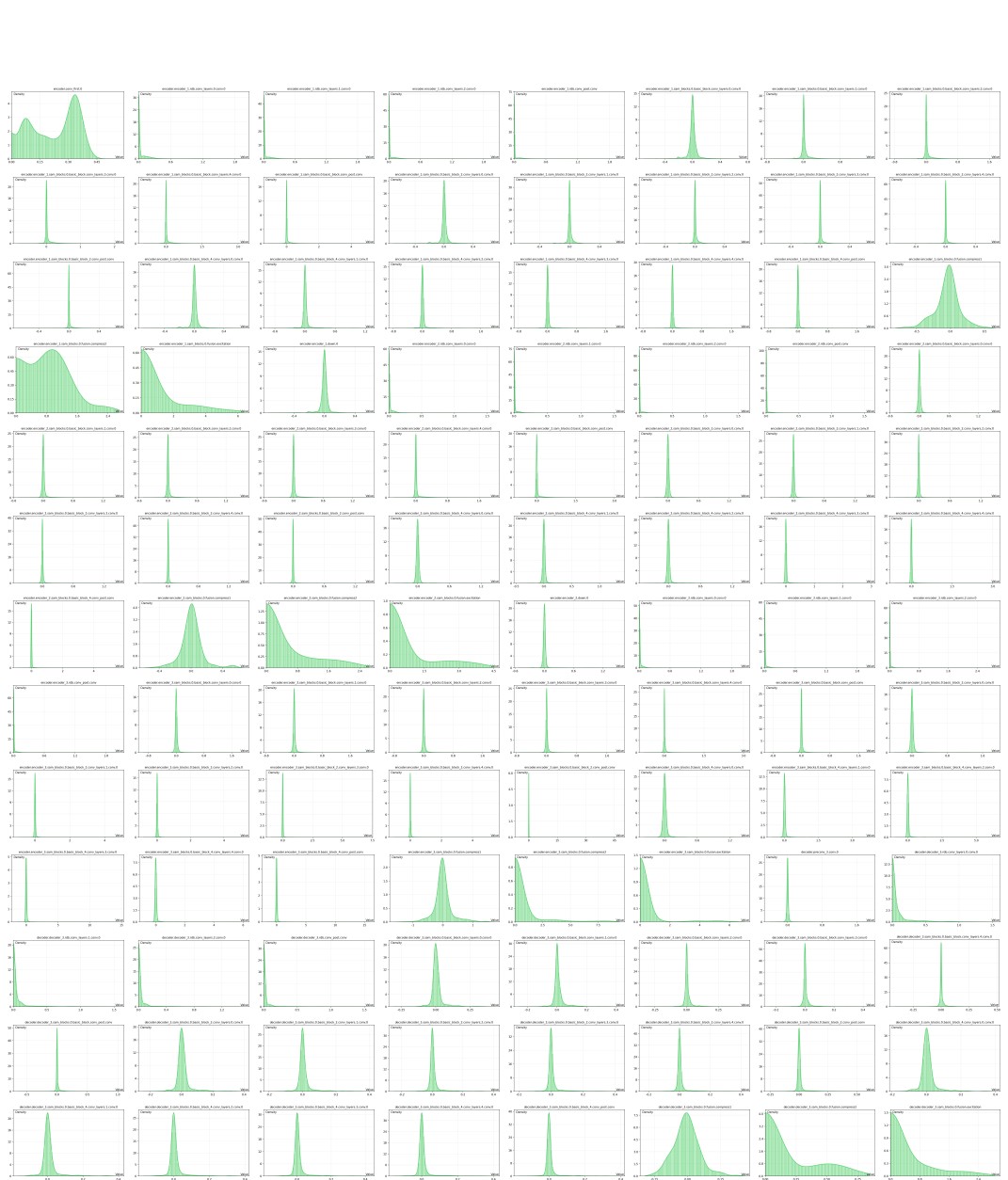

Figure 4: More distribution of activation (Part 1).

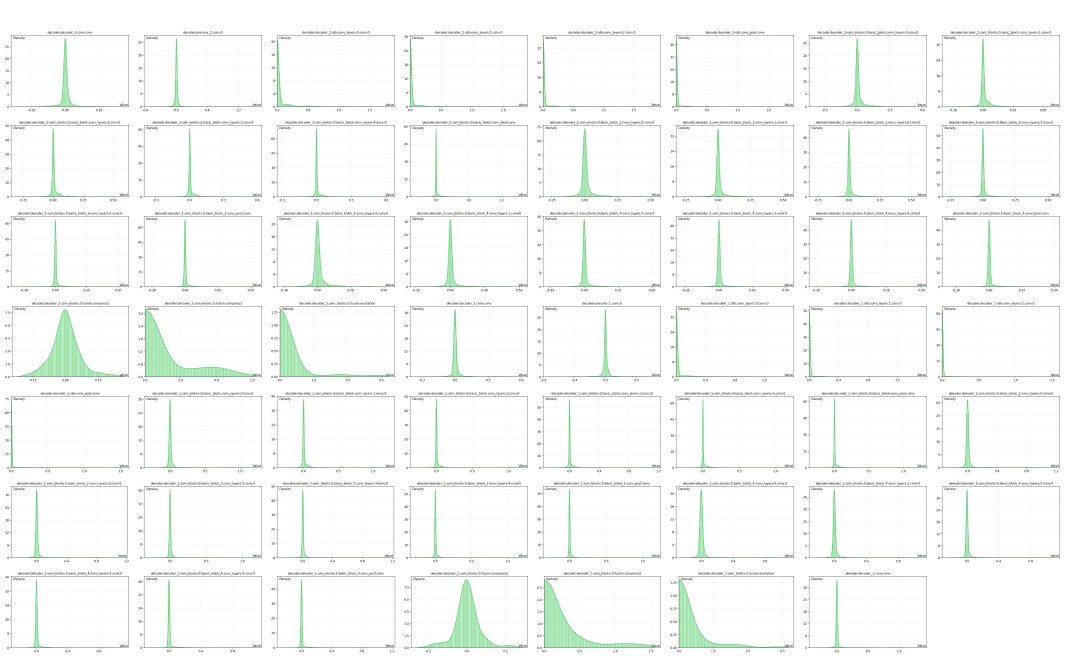

Figure 5: More distribution of activation (Part 2).

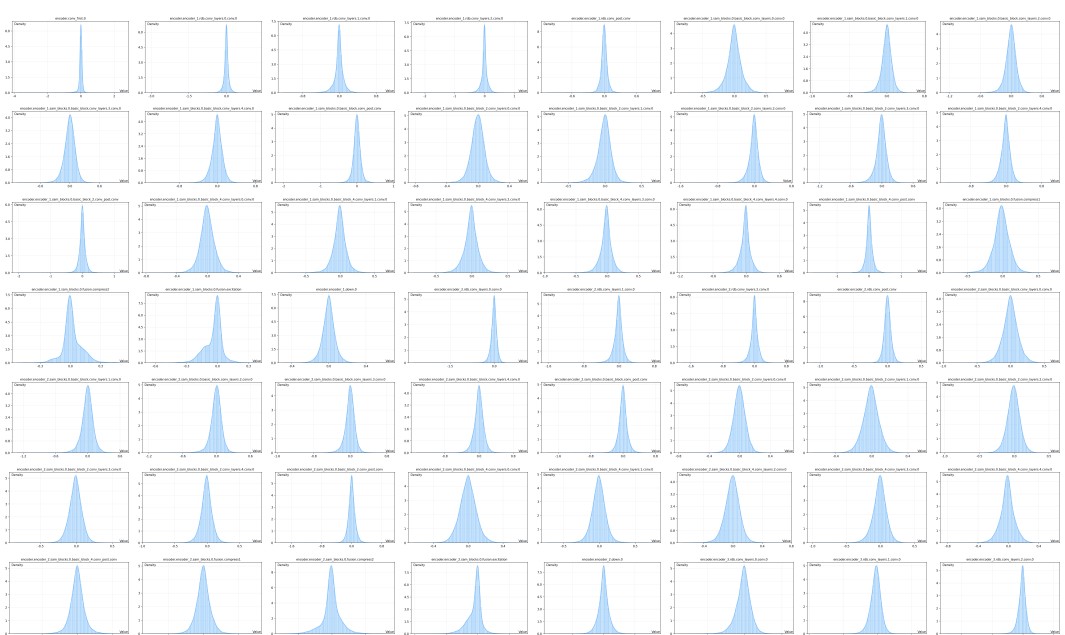

Figure 6: More distribution of weight (Part 1).

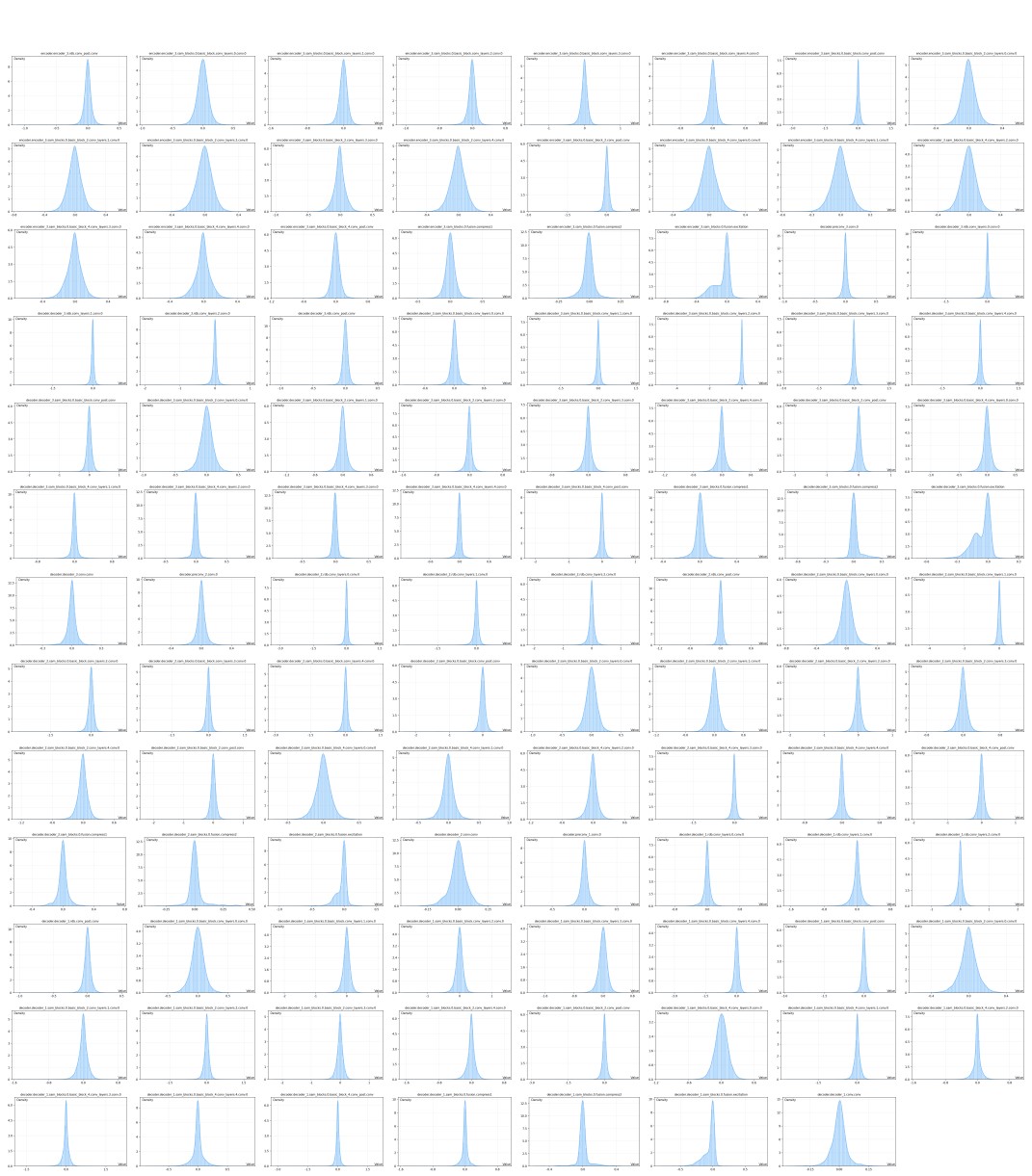

Figure 7: More distribution of weight (Part 2).