# OpenReview forum: "QuantDemoire: Quantization with Outlier Aware for Image Demoiréing"
_ICLR.cc/2026/Conference — Submitted to ICLR 2026_

### Official Review · Reviewer_H9oE · 2025-10-14

**Soundness:** 1
**Presentation:** 2
**Contribution:** 1
**Rating:** 2
**Confidence:** 4

**Summary:**

The authors aim to improve quantization for ML-based demoireing methods on the edge. Their method consists of three components:
1. setting the activation quantizer grid through a minmax of a *sample* of the activations (thereby reducing the quantization range, as some outliers may not be sampled),
2. keeping the worst outlier entries of the weights in Float16,
3. calibrating the activation quantization boundaries (i.e. clipping) based on some "frequency-aware" reconstruction loss

**Strengths:**

1. The authors address a real-world application, demoireing on the edge.
2. The ablations are quite extensive (but see my questions)
3. The results suggest the method performs favorably compared to existing works.

**Weaknesses:**

**1. Method**

Unfortunately, at this point I do not recommend acceptance. I believe there are a few issues with the method that I urge the authors to seriously consider:

**a. Activation Quantization**. The idea behind Eq. 6, randomly sampling activation entries to avoid outliers, is not effective---despite the authors' claim "the proposed method effectively captures the typical distribution of activations" and "[this means we] naturally discard extreme outliers". Just subsampling the entries does not change the distribution that is sampled from, it only changes the number of samples that is used for estimating the max value. Moreover, there is a large variance in this procedure---sometimes bad outliers may be sampled, and sometimes not. For example, it is not clear why Figure 3 right should be preferred---the real outliers are actually worse here. It would be much better in my opinion to compute e.g. the 95% quantiles for setting the range---this is not cheap, but even an approximation would suffice (e.g. `x>mean+std*some_constant`). It would be even better to learn how much to clip on some calibration data, which doesn't need to be expensive.

**b. Weight quantization**. The mixed-precision scheme makes sense, but because the mixed-precision is completely unstructured, the matmul would not be parallelizable (e.g. not run on GPU/TPU/NPU). The authors do not address this. Using mixed-precision for weights is also not new, so in any case this cannot really be claimed as a contribution.

**c. Calibration stage**. The authors' use of "quantization boundary optimization" (Section 3.3) is not clear to me. In 3.2, they propose the sampled activation quantizer approach, but now they say they optimize the boundary anyway.


**2. Results**

**a. stds**. The ablations do not have standard deviations. I assume these experiments are quite cheap, and hence std's would be feasible. They are valuable, because the sampling strategy of the activation quantization clearly has a high variance. The calibration approach also has some non-ignorable noise.

**b. Compression Ratio**. What do you mean with OPS? I assume the number of ops is the same (or slightly higher), but that the ops are cheaper in low-bit. In any case, these results should in my opinion be compared to Bfloat16, not Float32. Additionally, the effect of switching from INT to Float for some weight indices should not be underestimated and is not addressed by these results.

**c. Experimental details**. More experimental details would be desirable, see my two questions below.


## Minor

> [L.136] Although QAT achieves competitive performance (Nagel et al., 2021), its high training cost makes
it less suitable for deployment on edge devices.

Since training does not happen on the edge, this argument does not hold.

> [L. 213] Some existing post-training quantization methods (such as percentile (Li et al., 2019)) are specifically
designed to address the problems caused by outliers. However, these methods could lead to some
additional time overhead during the calibration stage. At the same time, because of their inflexible
design, these methods may lack robustness facing outliers in different distributions.

This could be explained better. Again it seems to me that calibration can happen not on the edge, so this is no issue. It is also not clear what is meant by the last sentence, "may lack robustness [due to their "inflexible design"]"

**Questions:**

1. Do all ablations use calibration of activation boundaries? E.g. also the "Original" and "Smooth (Raw)" baselines?

2. Can you explain why the percentile approach performs worse than your sampled approach?

---

> ### Author Response · Authors · 2025-11-24
> **Response to Reviewer H9oE (denoted as R3) part 1**
>
> ``Q3-1`` **Activation Quantization**. The idea behind Eq. 6, randomly sampling activation entries to avoid outliers, is not effective---despite the authors' claim "the proposed method effectively captures the typical distribution of activations" and "[this means we] naturally discard extreme outliers". Just subsampling the entries does not change the distribution that is sampled from, it only changes the number of samples that is used for estimating the max value. Moreover, there is a large variance in this procedure---sometimes bad outliers may be sampled, and sometimes not. For example, it is not clear why Figure 3 right should be preferred---the real outliers are actually worse here. It would be much better in my opinion to compute e.g. the 95% quantiles for setting the range---this is not cheap, but even an approximation would suffice (e.g. `x>mean+std*some_constant`). It would be even better to learn how much to clip on some calibration data, which doesn't need to be expensive.
>
> ``A3-1`` Thank you for the comment.
>
> **(1) effectiveness of random sampling **  Because outliers constitute only a minute fraction of the data, random subsampling substantially **reduces the probablity of including extreme outliers.** As a result, subsampling can accurately characterize the bulk of the activation distribution while mitigating the influence of rare extremes. The relevant **mathematical proofs** will be presented below.
>
> **(2) variance of randomly sampling** Given that extreme outliers comprise only a vanishingly small proportion of the data, the probability that random sampling includes them is negligible. Indeed, our method exhibits minimal variance and stable performance; the corresponding experimental results are presented below：
>
> | Run  |  PSNR   |  SSIM  | LPIPS  |
> | :--: | :-----: | :----: | :----: |
> |  1   | 21.0808 | 0.7626 | 0.3068 |
> |  2   | 21.0835 | 0.7623 | 0.3066 |
> |  3   | 21.0798 | 0.7628 | 0.3070 |
> |  4   | 21.0801 | 0.7624 | 0.3059 |
> |  5   | 21.0824 | 0.7631 | 0.3062 |
> | std  | 0.0014  | 0.0003 | 0.0004 |
>
> **(3) mathematical analysis of (1) & (2)**
>
>
>
> Supplementary Figures 4 and 5 indicate that, after smoothing, the activation distributions of almost all convolutional layers are well approximated by a Gaussian. Accordingly, we model the N entries of an activation tensor as $N$ independent, identically distributed standard normal variables.   As specified in Section 4.1 (Experimental Settings),  we adopt a sampling ratio of $\gamma = 0.1\%$, resulting in a sample size of $n = \gamma N$. Formally, let $X_1, \ldots, X_N \overset{\text{i.i.d.}}{\sim} \mathcal{N}(0,1)$ and denote the maximum of all values by $M_N = \max_{1 \leq i \leq N} X_i$ whose PDF is given by:
>
> $ f_{M_N}(x) = N\phi(x)\Phi(x)^{N-1},$
>
> where $\phi$ and $\Phi$ are the standard normal PDF and CDF. Then:
>
> $ \mathbb{E}[M_N] = \int_{-\infty}^{\infty} xN\phi(x)\Phi(x)^{N-1}dx.$
>
> Similarly, the expected maximum of the $n$ randomly selected activation values is: $\mathbb{E}[M_n] = \int_{-\infty}^{\infty} xn\phi(x)\Phi(x)^{n-1}dx.$
>
> This viewpoint lets us **interpret the sampling-based estimator as effectively clipping at the percentile $p$** determined by the CDF: $p = \Phi(\mathbb{E}[M_n])$. We observe that as the total number of elements ($N$) increases, the sampling-based estimation corresponds to a higher equivalent percentile threshold.  For **$N=10^6$, $p \approx 99.95\%$; for $N=10^7$, $p \approx 99.995 \%$; and for $N=10^8$, $p \approx 99.9995\%$**.  Moreover, due to smoothing, the count of extreme outliers tends to plateau for very large $N (N ≥ 10^6)$ rather than scale linearly, as shown in the table below:
>
> |   N    | \|x-mean\| > 4 * std | \|x-mean\| > 4.5 * std |
> | :----: | :------------------: | :--------------------: |
> | $10^6$ |          51          |           17           |
> | $10^7$ |         103          |           41           |
> | $10^8$ |         253          |           72           |
>
>  Therefore, by **adaptively clipping at higher percentiles for larger $N$**, our sampling-based estimation effectively removes extreme outliers while avoiding the pitfall of fixed-percentile clipping, which may incorrectly truncate non-extreme outlier values.
>
> **(4) explanation of Figure 3.** The purpose of the smoothing operation is to **align per‑channel activation magnitude**. In Figure 3(a), substantial inter‑channel magnitude discrepancies remain, and many outliers are present. In Figure 3(c), although the absolute magnitude of outliers increases, **their number decreases markedly, and the magnitude across channels becomes tightly aligned**, which reveals that random sampling more effectively achieves the objective of the smoothing operation.

---

> ### Author Response · Authors · 2025-11-24
> **Response to Reviewer H9oE (denoted as R3) part 2**
>
> **(5) experimental comparison between sampling and fixed‑percentile clipping** We present additional experiments demonstrating that our sampling method outperforms fixed‑percentile clipping. Specifically, we compare sampling with 95th‑ and 99.9th‑percentile clipping.  The results show that **sampling provides clear advantages over fixed‑percentile clipping** (Apart from using sampling-based estimation, 95th-percentile clipping, and 99.9% clipping respectively to compute the quantization boundaries and the smoothing factor, the remaining settings for all three methods are identical to the final configuration of our method.):
>
> |      method      |         PSNR         |        SSIM         |        LPIPS        |
> | :--------------: | :------------------: | :-----------------: | :-----------------: |
> | Ours (sampling)  | 20.2803 $\pm 0.0021$ | 0.6745 $\pm 0.0005$ | 0.3369 $\pm 0.0005$ |
> |  percentile 95%  |       19.3058        |       0.6026        |       0.3623        |
> | percentile 99.9% |       19.3326        |       0.6091        |       0.3487        |
>
> **(6)** **Why we do not learn the smoothing factor**  In fact, we train the quantization bounds during the calibration stage.  **(3)** has shown that, with a fixed number of activation values, sampling is equivalent to percentile clipping, so this effectively amounts to training the clipping percentile. However, **we do not train the smoothing factor because training quantization parameters is unstable due to the rounding operation**, and only training the quantization bounds yields better performance, as shown by the results below. Moreover, adding more learnable parameters increases the number of training epochs from 4 to 40, raising the training cost.
>
> ``Q3-2`` **b. Weight quantization**. The mixed-precision scheme makes sense, but because the mixed-precision is completely unstructured, the matmul would not be parallelizable (e.g. not run on GPU/TPU/NPU). The authors do not address this. Using mixed-precision for weights is also not new, so in any case this cannot really be claimed as a contribution.
>
> ``A3-2`` Thank you for the comment.
>
> We agree that naïvely preserving arbitrary weights in FP can break hardware-friendly parallelism. Our design keeps the dense path intact: the bulk of computation runs as standard INT8 convolutions on tensor cores, while **a small FP16 “outlier” branch operates at the element level with a dedicated kernel.** This avoids converting convolutions into unstructured sparse GEMMs, preserves coalesced memory access, and keeps the main matmul highly parallelizable on GPU/TPU/NPU backends. Because the FP16 fraction is very small, the extra kernel adds only 7% latency and 6% GPU memory in our measurements.
>
> Mixed-precision for weights is a common quantization strategy. For example, LLM.int8 processes a very small number of outlier feature dimensions in FP16, and SqueezeLLM stores a small subset of weights in a sparse format. Our contribution is independent of these methods and targets a different aspect: **a convolution-specific, outlier-aware kernel that isolates FP16 processing at element granularity without sparse reformatting.**  We employ CSR-style per-channel indexing for sparse weight storage. The computation strategy tiles output rows across CUDA blocks and stripes threads along the column dimension, with each kernel serially traversing the sparse weights and accumulating contributions to the output tensor. This design choice is justified by the extreme sparsity regime: for a typical 32×16×3×3 kernel, each output channel retains only 1-2 weight elements on average, making serial traversal more efficient than parallel scheduling overhead.  **Our implementation differs substantially from prevailing sparse‑convolution methods.**

---

> > ### Comment · Reviewer_H9oE · 2025-11-26
> >
> > Thank you for the response.
> >
> > Q1. I appreciate the multiple runs and extended discussion, but theoretically I still struggle; the variance of taking the maximum is large, especially for distributions with heavy tails, hence even the *sample standard deviation of your method will have large variance* and 5 runs would not be enough. Your theoretical analysis assumes a normal distribution, but this goes against the core of the paper; that quantization is hard **because** the distribution is heavy-tailed (what Figure 4 also shows). Above all, I just don't understand why this method should be preferred over a different robust estimator that does not have this variance issue, e.g. a 99.99% percentile estimator or an estimator that only takes mean+-k*std (e.g. k=5). Just to make my concern very explicit, e.g. see toy example
> > ```
> > import torch
> > torch.manual_seed(0)
> >
> > n = 10000
> > n_runs = 5
> > n_samples = 5
> > prop_sampled = 0.1
> > for _ in range(n_runs):
> >     maxes = torch.empty(n_samples)
> >     for subrun in range(n_samples):
> >         x = torch.randn(n)
> >         x[0] = 100 # <- outlier
> >         subset = torch.rand(n) < prop_sampled
> >         max_ = torch.max(x[subset])
> >         maxes[subrun] = max_
> >     print(torch.std(maxes).item())
> >
> >
> > >>>
> > 0.2660759687423706
> > 43.199764251708984
> > 0.3908720314502716
> > 0.6617917418479919
> > 0.33608126640319824
> > ```
> > so some runs (run 2 in this case) will yield a completely different estimate due to it sampling the outlier.
> >
> > I appreciate the experimental comparison against 99.9% percentile. From your analysis earlier, it seems a much higher percentile might be needed, e.g. 99.99% or 99.999%, to give a better comparison. This would be a single hyperparameter, just like your number of samples is also a hyperparameter.
> >
> > At last,
> > > we do not train the smoothing factor because training quantization parameters is unstable due to the rounding operation
> > this is not in line with related work, which does do this successfully (e.g. FlatQuant).
> >
> > Q2. Thanks! If your kernel is much better than the existing work's, I would present it more prominently as a contribution.

---

> > > ### Comment · Reviewer_H9oE · 2025-11-26
> > >
> > > Q3. Thanks, that makes sense. Have you tried training for longer/changing the training LR and such to improve it further? From my experience, initialization helps, but your claim `training alone can only reduce quantization error within a small range` is not true for longer training.
> > >
> > > Q4. Do you agree this analysis is not valid, considering your method is designed explicitly for non-Gaussian outliers? See also Q1
> > >
> > > Q5. Thanks. Please explain this in the paper as otherwise a "80%+ reduction" is meaningless. Why would your baseline FP32 and FP16/BF16, considering you show that FP16 is good enough to capture outliers? Additionally, you mention in the revised paper that the memory and inference overhead in the paper, of 6 and 7%. Can you give details on this?
> > >
> > > Q6. Thanks!

---

> ### Author Response · Authors · 2025-11-24
> **Response to Reviewer H9oE (denoted as R3) part 3**
>
> ``Q3-3`` Calibration stage. The authors' use of "quantization boundary optimization" (Section 3.3) is not clear to me. In 3.2, they propose the sampled activation quantizer approach, but now they say they optimize the boundary anyway.
>
> ``A3-3`` Thank you for the comment.
>
> The sampled-activation quantizer approach aims to **initialize the quantization boundary to a favorable starting point**, whereas boundary **optimization subsequently performs fine-grained refinement**. Because rounding operations in quantization introduce non-differentiability and can destabilize training, **training alone can only reduce quantization error within a small range**; a well-chosen initialization of the quantization boundary remains crucial. The following experiments substantiate this claim.  We initialize the quantization boundaries using different methods, keep all other settings the same, and train each method for a sufficient number of epochs. The results show that initializing the quantization boundaries with sampling-based estimation still offers a clear advantage.
>
> |      method      |         PSNR         |        SSIM         |        LPIPS        |
> | :--------------: | :------------------: | :-----------------: | :-----------------: |
> | Ours (sampling)  | 20.2803 $\pm 0.0021$ | 0.6745 $\pm 0.0005$ | 0.3369 $\pm 0.0005$ |
> |  percentile 95%  |       19.4556        |       0.6223        |       0.3601        |
> | percentile 99.9% |       19.5781        |       0.6391        |       0.3532        |
> |      MinMax      |       18.9265        |       0.5921        |       0.4013        |
>
> This two-stage paradigm—good initialization followed by fine-grained refinement—is already established in the literature, e.g., in 2DQuant[4].
>
> > [4] Liu, Kai et al. “2DQuant: Low-bit Post-Training Quantization for Image Super-Resolution.” *ArXiv* abs/2406.06649 (2024): n. pag.
>
> ``Q3-4`` **a. stds**. The ablations do not have standard deviations. I assume these experiments are quite cheap, and hence std's would be feasible. They are valuable, because the sampling strategy of the activation quantization clearly has a high variance. The calibration approach also has some non-ignorable noise.
>
> ``A3-4`` Thank you for the comment.
>
> Because extreme outliers are exceedingly rare and the number of activation elements is large, the sampling procedure does not incur high variance and is in fact quite stable. Under the 4-bit UHDM setting, we ran our final configuration five independent times; the resulting standard deviation is negligible relative to the magnitude of the evaluation metric. We also provide a theoretical justification. Continuing the derivation in **A1-1 (3)**, and using a first-order Taylor expansion (Delta method), we obtain
>
> $\mathrm{std}(p) = \mathrm{std}(\Phi(M_n)) \approx [\phi(\mathbb{E}[M_n])] \mathrm{std}(M_n)$
>
> where
>
> $\mathbb{E}[M_n^2] = \int_{-\infty}^{\infty} x^2 f_{M_n}(x) dx = \int_{-\infty}^{\infty} x^2 n \phi(x) \Phi(x)^{n-1} dx.$
>
> Numerical calculations for $N = 10^6,10^7$and  $10^8$ yield **standard deviations of 6.522e-04, 6.596e-05, and 6.653e-06**, respectively, which are  negligible.
>
> ``Q3-5`` **b. Compression Ratio**. What do you mean with OPS? I assume the number of ops is the same (or slightly higher), but that the ops are cheaper in low-bit. In any case, these results should in my opinion be compared to Bfloat16, not Float32. Additionally, the effect of switching from INT to Float for some weight indices should not be underestimated and is not addressed by these results.
>
> ``A3-5`` Thank you for the comment.
>
> OPS quantifies compute in **bit-operations**. Relative to an FP32 baseline, we assume linear scaling with bit width; thus, the cost of b‑bit arithmetic is estimated as $Ops_b = (b/32)·Ops_{fp32}$. We also account for the overhead introduced by FP16 weights, defined as $Ops_{w16} = Ops_{w32}/2$, where $Ops_{w32}$ denotes the number of bit-operations of  preserved outlier weights in float32. The final reported compute is $Ops_{total} = Ops_b + Ops_{w16}$.

---

> ### Author Response · Authors · 2025-11-24
> **Response to Reviewer H9oE (denoted as R3) part 4**
>
> ``Q3-6`` Do all ablations use calibration of activation boundaries? E.g. also the "Original" and "Smooth (Raw)" baselines?
>
> ``A3-6`` Thank you for the comment.
>
> Here we present the full experimental settings for all ablation studies reported in the main text:
>
> All ablation experiments were conducted using static quantization.
>
> **Table 1 (a) Quantizer bound calculation:** All methods are evaluated without smoothing and without any training-based optimization. Weights are quantized using MinMax. For activation quantization: MinMax uses the absolute minimum and maximum; Percentile applies 99.9th-percentile clipping; Sample uses the sampling-based estimator proposed in this paper.
>
> **Table 1 (b) Smooth transformation:** In this setting, weights are quantized by MinMax, activations are quantized using the sampling-based estimator, and no training-based optimization is applied. Original applies no smoothing; Smooth (Raw) computes the smoothing factor from the absolute maximum activation; Smooth (Sample) computes the smoothing factor using the sampling-based estimator.
>
> **Table 2 (a) Outlier handling method** Baseline and Smooth (Sample) in Table 1 (b) share identical settings. Discard Outliers directly truncates weight outliers; Store Random randomly stores 1% of weights in FP16; Store Outlier stores the largest 0.5% and smallest 0.5% of weight elements.
>
> **Table 2 (b) Outliers Proportion ** 0.50% is identical to Store Outlier in Table 2 (a); 0.25% stores the largest 0.25% and smallest 0.25% of weight elements (all other settings identical to the 0.50% case); 1% stores the largest 1% and smallest 1% of weight elements (all other settings identical to the 0.50% case).**
>
> **Table 3 (a) Calibration strategy**  No Calibration uses the same settings as Store Outlier in Table 2 (a). Original Image applies $L_1$ & $L_p$​ calibration directly on the original images (on top of No Calibration). High-Freq performs calibration using high-frequency components of the images. Mid&Low-Freq performs calibration using mid- and low-frequency components.
>
> **Table 3 (b) Frequency extraction level**  For level = 3, the settings are identical to Mid&Low-Freq in Table 3 (a). For level = 1 and level = 5, we use 1 and 5 convolution kernels, respectively, to extract frequency components; all other settings are the same as for level = 3.

---

> ### Author Response · Authors · 2025-11-27
> **Response to Reviewer H9oE part 1**
>
> Thank you for your thoughtful comments. Our responses are listed below. We number items as FQ3‑X (e.g., FQ3‑1‑1, FQ3‑2, etc.).
>
> ``FQ3-1-1`` Your theoretical analysis assumes a normal distribution, but this goes against the core of the paper; that quantization is hard **because** the distribution is heavy-tailed (what Figure 4 also shows).
>
> ``FA3-1-1``
>
> Thank you for the comment.
>
> In our paper we claim that  "From Fig. 4, it can be observed that, most activations conform to either a Gaussian or an exponential distribution. Such distributions
> typically exhibit a small number of outliers that deviate from the normal range on one or both sides."  Therefore, the normality assumption aligns with the distribution of activation values.
>
> ``FQ3-1-2`` Above all, I just don't understand why this method should be preferred over a different robust estimator that does not have this variance issue, e.g. a 99.99% percentile estimator or an estimator that only takes mean+-k*std (e.g. k=5). So some runs (run 2 in this case) will yield a completely different estimate due to it sampling the outlier.
>
> ``FA3-1-2``
>
> Thank you for the comment.
>
> Your toy example is very valuable. However, several settings should be adjusted to better match the practical conditions of our method. First, $n$ should be much larger—for a 4K image, `n` is about $10^8$. Second, in our method we set `prop_sampled` to 0.001. Finally, rather than using the sample maximum, computing **percentile** is preferable because it more accurately targets the clipping rate. The revised code and corresponding results are as follows:
>
> ```
> import torch
> torch.manual_seed(0)
>
> n = 100000000
> n_runs = 5
> n_samples = 5
> prop_sampled = 0.001
>
> for _ in range(n_runs):
>     percentiles = torch.empty(n_samples)
>     for subrun in range(n_samples):
>         x = torch.randn(n)
>         x[0] = 100.0  # outlier
>         subset = torch.rand(n) < prop_sampled
>         while not subset.any():
>             subset = torch.rand(n) < prop_sampled
>
>         max_ = torch.max(x[subset])
>         pct = (x <= max_).float().mean() * 100.0
>         percentiles[subrun] = pct
>
>     mean = percentiles.mean().item()
>     std = percentiles.std(unbiased=True).item()
>     print(f"{mean:.4f} {std:.4f}")
>
> >>>
> 99.9993 0.0006
> 99.9991 0.0010
> 99.9990 0.0010
> 99.9980 0.0012
> 99.9994 0.0005
>
> ```
>
> We observe that the standard deviation is negligible relative to the mean percentage.
>
> ``FQ3-1-3``
>
> I appreciate the experimental comparison against 99.9% percentile. From your analysis earlier, it seems a much higher percentile might be needed, e.g. 99.99% or 99.999%, to give a better comparison. This would be a single hyperparameter, just like your number of samples is also a hyperparameter.
>
> ``FA3-1-3``
>
> Thank you for the comment.
>
> We evaluated a 99.999% threshold on two datasets. On FHDMi, this setting performs comparably to our method; on LCDMoire, however, the activations are fewer, so the 99.999%‑percentile fails to effectively clip outliers, giving our method a clear advantage.
>
> |       method       | Dataset  |  PSNR   |  SSIM  | LPIPS  |
> | :----------------: | :------: | :-----: | :----: | ------ |
> |  Ours (sampling)   |  FHDMi   | 20.2803 | 0.6745 | 0.3369 |
> | percentile 99.999% |  FHDMi   | 20.2856 | 0.6367 | 0.3576 |
> |  Ours (sampling)   | LCDMoire | 31.2830 | 0.9544 | 0.1793 |
> | percentile 99.999% | LCDMoire | 25.3156 | 0.9184 | 0.2313 |
>
> ``FQ3-1-4``
>
> we do not train the smoothing factor because training quantization parameters is unstable due to the rounding operation this is not in line with related work, which does do this successfully (e.g. FlatQuant).
>
> ``FA3-1-4``
>
> Thank you for the comment.
>
> FlatQuant indeed achieves strong performance when training the smoothing factor. We also adopted the OmniQuant strategy to jointly train the smoothing factor and the quantization bounds. As shown below, even with additional training epochs, this approach still slightly underperforms our method (we swept multiple learning rates and report the best result).
>
> |                         method                         |  PSNR   |  SSIM  | LPIPS  | training epoch |
> | :----------------------------------------------------: | :-----: | :----: | :----: | :------------: |
> |          Ours (only learn the clipping bound)          | 21.0808 | 0.7626 | 0.3068 |       4        |
> | Omniquant (learn clipping bound and smoothing factor ) | 21.0643 | 0.7613 | 0.3106 |       40       |
>
> This discrepancy may come from architectural differences between LLMs (Transformers) and demoireing networks (CNNs).
>
> ``FQ3-2`` Thanks! If your kernel is much better than the existing work's, I would present it more prominently as a contribution.
>
> ``FA3-2`` Thank you for recognizing the strength of our kernel. We will make its advantages over existing methods more visible in the revision.

---

> ### Author Response · Authors · 2025-11-27
> **Response to Reviewer H9oE part 2**
>
> ``FQ3-3`` Have you tried training for longer/changing the training LR and such to improve it further? From my experience, initialization helps, but your claim `training alone can only reduce quantization error within a small range` is not true for longer training.
>
> `FA3-3`
>
> Thank you for the comment.
>
> In the reported results, all methods were trained for a sufficient number of epochs. Our method converged within 4 epochs, whereas the other three methods showed no further improvement after 10 epochs. Other work, such as 2DQuant, offers a similar explanation: “DQC alone cannot match the optimization efficacy of DOBI. This is because the influence of quantizer parameters on model performance is oscillatory, and training alone tends to converge to local optima.”
>
> ``FQ3-4``  Do you agree this analysis is not valid, considering your method is designed explicitly for non-Gaussian outliers? See also Q1
>
> ``FA3-4``
>
> Thank you for the comment.
>
> In ``FA3-1-1``, we have already indicated that our paper observes the demoiréing network’s activations to follow either a normal or an exponential distribution, thereby supporting the validity of our analysis.
>
> ``FQ3-5``  Thanks. Please explain this in the paper as otherwise a "80%+ reduction" is meaningless. Why would your baseline FP32 and FP16/BF16, considering you show that FP16 is good enough to capture outliers? Additionally, you mention in the revised paper that the memory and inference overhead in the paper, of 6 and 7%. Can you give details on this?
>
> ``FA3-5``
>
> Thank you for the comment.
>
> Here, 80% denotes the theoretical reduction in the bit-width of both parameters and computations. We use FP32 as the baseline because the original ESDNet model was implemented in FP32; we therefore adopt the same setting for a fair comparison.
>
> To quantify the additional overhead introduced by the FP16 branch, we conduct two experiments. In the first, we execute a 64×64×3×3 convolution under W4A4 quantization (4-bit weights, 4-bit activations) and repeat it 50 times. In the second, we run the same workload under W4A4 but randomly route 1% of the weights through an FP16 branch. We record peak memory usage and wall-clock time for both settings. The W4A4-only baseline consumes 256.12 MB and 1493.42 ms, whereas the configuration with the FP16 branch consumes 271.49 MB and 1597.95 ms, corresponding to approximately a 6% memory overhead and a 7% runtime overhead.

---

### Official Review · Reviewer_FcUX · 2025-10-24

**Soundness:** 2
**Presentation:** 3
**Contribution:** 2
**Rating:** 2
**Confidence:** 3

**Summary:**

The paper introduces QuantDemoire: a post training quantization methodology designed specifically for the quantization of Image Demoiréing models. The methodology consist of an alternative sample-base range-setting methodology, which preserves a small percentage of weights in high precision, and a calibration method focused on preserving low and medium frequencies of the model output. An empirical evaluation demonstrates the effectiveness of the proposed methodology in extreme quantization settings, by comparing QuantDemoire against recent approaches developed in the context of image generation quantization literature.

**Strengths:**

* The paper introduces a novel method for effectively quantizing demoreing models, which is a novel application. The paper compares the demoreing performance against strong baselines used in Diffusion Transformer quantization.

* Overall, the paper does a good job in describing all the components. The methodology section is overall quite simple and clear.

**Weaknesses:**

* Recent edge devices can handle models of up to a few billion parameters (fp16/bf16) hence it should be possible to deploy models consisting of million of parameters without aggressive quantization. The motivation is not entirely obvious from reading sections 1 and 2.

* The sampling procedure described in section 3.2 seems like a high-variance estimate of the k-th percentile. However, the two methods are not compared theoretically, and the setting used for the empirical comparison are unclear. The computational cost of estimating distribution statistics is usually negligible since the calibration phase needs to be performed only once and yields lower variance estimate than the proposed sampling approach. Variance estimations are not reported in the experimental results in Table 1a.

* Figure 4 is difficult to read because of the low density of the distribution tails. Please consider using a logarithmic scale.

**Questions:**

1. Can the authors elaborate on the statement “[edge devices] are also the most important application scenarios [for image demoiréing]” ? Although the relevance of the task is clear, this statement does not seem entirely obvious. Deploying models consisting of millions of parameters on the edge with recent hardware should also be possible without aggressive quantization. What is the ideal use case of QuantDemoire?

2. What hyper-parameter ranges are compared to produce the results reported in Table 1a? What is the variance on the proposed sampling methodology when compared to the baseline? Can the authors further comment on the advantages of sampling vs percentile-based range-setting?

3. The paper proposes to keep the weight outliers in full precision instead of clipping them. Can the authors quantify the overhead introduced by $W_{outlier}$ matmul in terms of memory and latency? Is the bit width substantially lower than a less-aggressive quantization scheme (e.g. 5 bits) ? How is the sparse matmul performed efficiently?

4. What is the effect of the two terms $\mathcal{L}_1$ and $\mathcal{L}_p$ on the loss? Is there a tunable weighting factor? I suspect that, due to the different number of dimensions the two terms might have a different scale.

---

> ### Author Response · Authors · 2025-11-24
> **Response to Reviewer FcUX (denoted as R2) part 1**
>
> ``Q2-1`` Recent edge devices can handle models of up to a few billion parameters (fp16/bf16) hence it should be possible to deploy models consisting of million of parameters without aggressive quantization. The motivation is not entirely obvious from reading sections 1 and 2.
>
> ``A2-1`` Thank you for the comment.
>
> Deploying models with millions of parameters on modern hardware such as GPUs is straightforward. However, demoiréing models may need to operate on **real-time image/video processing devices**, including DSLR cameras, drones, and IoT sensors. These devices are characterized by **severely limited computational power** and stringent low-latency requirements. Therefore, aggressive quantization of demoiréing models is essential for these application scenarios.
>
> ``Q2-2`` The sampling procedure described in section 3.2 seems like a high-variance estimate of the k-th percentile. However, the two methods are not compared theoretically, and the setting used for the empirical comparison are unclear. The computational cost of estimating distribution statistics is usually negligible since the calibration phase needs to be performed only once and yields lower variance estimate than the proposed sampling approach. Variance estimations are not reported in the experimental results in Table 1a.
>
> ``A2-2`` Thank you for the comment.
>
> We demonstrate both **experimentally** and **theoretically** that our sampling procedure achieves superior performance over fixed-percentile clipping and introduces negligible variance.
>
> **(a) Experimental results**
>
> As shown in Table 1a, our sampling procedure **surpasses the performance of fixed-percentile clipping**.
>
> To validate the stability of our method, we assessed the variance by conducting five  runs.
>
> | Run  |  PSNR   |  SSIM  | LPIPS  |
> | :--: | :-----: | :----: | :----: |
> |  1   | 21.0808 | 0.7626 | 0.3068 |
> |  2   | 21.0835 | 0.7623 | 0.3066 |
> |  3   | 21.0798 | 0.7628 | 0.3070 |
> |  4   | 21.0801 | 0.7624 | 0.3059 |
> |  5   | 21.0824 | 0.7631 | 0.3062 |
> | std  | 0.0014  | 0.0003 | 0.0004 |
>
> The **standard deviation** of each metric across five runs is negligible, demonstrating the **stability** of our method.
>
> The experimental configuration in Table 1a:  weights for both methods were quantized to 4-bit precision using a min-max scheme, with no smoothing or training-based calibration. **The sole variable** was the clipping method: our sampling procedure utilized the outlier-aware quantizer's sampling technique to determine the quantization bound, while the baseline percentile method used a fixed 99.9% threshold.

---

> ### Author Response · Authors · 2025-11-24
> **Response to Reviewer FcUX (denoted as R2) part 2**
>
> **(b)** **Theoretical Analysis**
>
> As illustrated in Figures 4 and 5 of the supplementary material, the activation distributions of nearly all convolutional layers approximate a normal distribution following the smoothing operation.  Consequently, we model the $N$ elements within a given activation tensor as $N$ independent and identically distributed (i.i.d.) random variables drawn from a standard normal distribution.  As described in Section 4.1 EXPERIMENTAL SETTINGS, we use a sampling ratio of $\gamma = 0.1\%$, resulting in a sample size of $n = \gamma N$. Formally, let $X_1, \ldots, X_N \overset{\text{i.i.d.}}{\sim} \mathcal{N}(0,1)$ and denote the maximum of all values by $M_N = \max_{1 \leq i \leq N} X_i$. Its probability density function (PDF) is given by:
>
> $ f_{M_N}(x) = N\phi(x)\Phi(x)^{N-1},$
>
> where $\phi$ and $\Phi$ are the standard normal PDF and cumulative distribution function (CDF), respectively. The expectation is then:
>
> $ \mathbb{E}[M_N] = \int_{-\infty}^{\infty} xN\phi(x)\Phi(x)^{N-1}dx.$
>
> Similarly, the expected maximum of the $n$ randomly selected activation values is: $\mathbb{E}[M_n] = \int_{-\infty}^{\infty} xn\phi(x)\Phi(x)^{n-1}dx.$
>
> This allows us to **treat the sampling-based estimation as being equivalent to clipping at a percentile threshold**, $p$, for our sampling-based estimation using the CDF: $p = \Phi(\mathbb{E}[M_n])$. **We observe that as the total number of elements ($N$) increases, the sampling-based estimation corresponds to a higher equivalent percentile threshold.  For instance, for $N=10^6$, $p \approx 99.95\%$; for $N=10^7$, $p \approx 99.995\%$; and for $N=10^8$, $p \approx 99.9995\%$.**  Furthermore, we find that due to the smoothing operation, the number of extreme outliers tends to stabilize for very large $N$ ($N > 10^6$) rather than growing linearly, as shown in the table below:
>
> |   N    | \|x-mean\| > 4 * std | \|x-mean\| > 4.5 * std |
> | :----: | :------------------: | :--------------------: |
> | $10^6$ |          51          |           17           |
> | $10^7$ |         103          |           41           |
> | $10^8$ |         253          |           72           |
>
> Where |x-mean| > 4 * std means the number of activations that lie more than four standard deviations away from the mean, and |x-mean| > 4.5 * std has a similar meaning.
>
> Therefore, by dynamically clipping at a higher percentile for larger $N$​, our sampling-based estimation effectively removes extreme outliers while avoiding the pitfall of fixed-percentile clipping, which may incorrectly truncate non-outlier values.
>
> Based on this analysis, we proceed to calculate the standard deviation of  $p$. Using a first-order Taylor expansion (the Delta Method), the standard deviation can be approximated as:
>
> $\mathrm{std}(p) = \mathrm{std}(\Phi(M_n)) \approx [\phi(\mathbb{E}[M_n])] \mathrm{std}(M_n)$
>
> where
>
> $\mathbb{E}[M_n^2] = \int_{-\infty}^{\infty} x^2 f_{M_n}(x) dx = \int_{-\infty}^{\infty} x^2 n \phi(x) \Phi(x)^{n-1} dx$
>
> Numerical calculations for $N = 10^6,10^7$and  $10^8$ **yield standard deviations of 6.522e-04, 6.596e-05, and 6.653e-06**, respectively, which are  negligible.
>
> ``Q2-3`` Figure 4 is difficult to read because of the low density of the distribution tails. Please consider using a logarithmic scale.
>
> ``A2-3`` Thank you for the comment.
>
> To highlight that outliers are sparse and lie far from the majority of the distribution, we plot the figures in a way that retains relatively long tails. To reduce unused whitespace, we will consider using a logarithmic scale.
>
> ``Q2-4`` Can the authors elaborate on the statement “[edge devices] are also the most important application scenarios [for image demoiréing]” ? Although the relevance of the task is clear, this statement does not seem entirely obvious. Deploying models consisting of millions of parameters on the edge with recent hardware should also be possible without aggressive quantization. What is the ideal use case of QuantDemoire?
>
> ``A2-4`` Thank you for the comment.
>
> While high-end edge devices can indeed run large models, our work is motivated by the most **resource-constrained** yet critical platforms for demoiréing. These include embedded systems like DSLR cameras, drones, and various IoT sensors, which operate under **stringent computational, power, and low-latency constraints**. For such real-time applications, aggressive quantization is not merely an optimization but is often essential for on-device deployment. The ideal use case for QuantDemoire is therefore enabling high-quality demoiréing on these computationally limited platforms.

---

> ### Author Response · Authors · 2025-11-24
> **Response to Reviewer FcUX (denoted as R2) part 3**
>
> ``Q2-5`` What hyper-parameter ranges are compared to produce the results reported in Table 1a? What is the variance on the proposed sampling methodology when compared to the baseline? Can the authors further comment on the advantages of sampling vs percentile-based range-setting?
>
> ``A2-5`` Thank you for the comment.
>
> Could you please explain "What hyper-parameter ranges are compared to produce the results reported in Table 1a?"?  The variance of the sampling methodology and the analysis on advantages of sampling are demonstrated in ``A2-2``.
>
> ``Q2-6`` The paper proposes to keep the weight outliers in full precision instead of clipping them. Can the authors quantify the overhead introduced by matmul in terms of memory and latency? Is the bit width substantially lower than a less-aggressive quantization scheme (e.g. 5 bits) ? How is the sparse matmul performed efficiently?
>
> ``A2-6`` Thank you for the comment.
>
> The FP16 branch for outlier weights incurs a minor overhead, adding **7% time and 6% to GPU memory**. However, since this branch stores only 1% of the total weights, the effective bit-width is just 4.12, which remains significantly lower than 5 bits.
>
> We employ CSR-style per-channel indexing for sparse weight storage. The computation strategy tiles output rows across CUDA blocks and stripes threads along the column dimension, with each kernel serially traversing the sparse weights and accumulating contributions to the output tensor. This design choice is justified by the extreme sparsity regime: for a typical 32×16×3×3 kernel, each output channel retains only 1-2 weight elements on average, making serial traversal more efficient than parallel scheduling overhead.
>
> ``Q2-7`` What is the effect of the two terms $\mathcal{L}_1$ and $\mathcal{L}_p$ on the loss? Is there a tunable weighting factor? I suspect that, due to the different number of dimensions the two terms might have a different scale.
>
> ``A2-7`` Thank you for the comment.
>
> Our loss function combines a pixel-wise $\mathcal{L}_1$ term and a perceptual loss term, $\mathcal{L}_p$. We do not propose a novel loss formulation; instead, consistent with prior demoiréing methods such as ESDNet and Freqformer, we simply set the weight for **both terms to 1**.

---

> ### Comment · Reviewer_FcUX · 2025-11-26
>
> I wish to thank the authors for the detailed answers and for clarifying the relevance of their work, which makes me appreciate the applicability of their method to real-life use-cases.
>
> Regarding the theoretical analysis, I appreciate the clarification and the additional experimental results on the standard deviation of the estimate.  There is one aspect I still find somewhat unclear, related to the demonstrated equivalence between the sampling-based and percentile threshold estimation. Specifically, could the authors elaborate on the statement:"Therefore, by dynamically clipping at a higher percentile for larger $N$​, our sampling-based estimation effectively removes extreme outliers while avoiding the pitfall of fixed-percentile clipping, which may incorrectly truncate non-outlier values." ?
>
> Given a finite set of samples $\mathbf{X} = \\{x_1,..., x_n\\}$, wouldn’t the percentile-based and sampling-based approaches yield similar clipping ranges under appropriate parameter choices (e.g.,$percentile(\mathbf{X}, p=0.995)\approx sample(\mathbf{X}, N=10^7)$? If so, why does the estimation method matter? What “dynamic” aspects are being referred to here?
> Does the difference lie in the clipping range being determined dynamically at inference time rather than statically from a finite calibration set, as is typical with percentile-based estimates?
>
> To clarify, my intention is not to question the effectiveness of the sampling-based approach, but rather to understand what fundamental novelty it introduces compared to standard percentile clipping.

---

> ### Author Response · Authors · 2025-11-27
> **Response to Reviewer FcUX**
>
> Thank you for your thoughtful comments. Our responses are listed below numbered ``FQ2-1`` and ``FQ2-2``.
>
> ``FQ2-1`` Specifically, could the authors elaborate on the statement:"Therefore, by dynamically clipping at a higher percentile for larger , our sampling-based estimation effectively removes extreme outliers while avoiding the pitfall of fixed-percentile clipping, which may incorrectly truncate non-outlier values." ?
>
> ``FA2-1``
>
> Thank you for the comment.
>
> In ``A2-2`` (b) , we prove that as the total number of activation elements increases, the sampling-based estimation corresponds to a higher equivalent percentile. This means that, **as the number of activation elements increases, the fraction of elements we clip should progressively decrease.** The table in  ``A2-2`` (b) shows that, in the demoiréing network, when the number of activation elements is already very large, further increases lead to an even smaller proportion of extreme outliers; hence, the clipping fraction should indeed continue to decline. In contrast, fixed-percentile clipping keeps the clipped fraction constant as the number of activation elements grows and may therefore truncate non-extreme values.
>
> ``FQ2-2`` If so, why does the estimation method matter? What “dynamic” aspects are being referred to here? Does the difference lie in the clipping range being determined dynamically at inference time rather than statically from a finite calibration set, as is typical with percentile-based estimates?
>
> ``FA2-2``
>
> Thank you for the comment.
>
> In demoiréing networks, operations such as up- and down-sampling lead to different numbers of activation elements across layers. Consequently, when the activation sampling ratio (our method’s only hyperparameter) is fixed, **different layers will have different pruning ratios**; this is what we mean by **"dynamic."** In contrast, the fixed-percentage baseline applies **the same clipping percentage to all layers.** Our method employs static quantization to reduce inference-time overhead.

---

### Official Review · Reviewer_piwj · 2025-10-30

**Soundness:** 3
**Presentation:** 3
**Contribution:** 2
**Rating:** 4
**Confidence:** 3

**Summary:**

Image demoireing is an important vision task especially for edge devices such as smartphones, drones, and portable cameras.
Thus, model quantization is necessary to utilize high performance demoireing models in practical applications.
Existing quantization methods are not focused on demoireing so that they 1) overlook outliers in weights and activations, and 2) weaken representations in smooth regions, resulting in a huge performance loss.
QuantDemoire removes activation outliers through random sampling and keeps extreme weights in FP to mitigate the impact of outliers.
It also extracts mid- and low-frequency information by a recursive kernel to train the quantized model to preserve low-frequency features well.
Extensive experiments show that QuantDemoire outperforms existing methods.

**Strengths:**

* The work successfully identifies and addresses the frequency-related issue that is unique to the demoireing task.
* Effectively extracting low-frequency features through a simple convolution kernel.
* QuantDemoire achieves the state-of-the-art performance.

**Weaknesses:**

* Smoothing and clipping approach of outlier-aware quantizer strongly resembles existing methods such as OmniQuant [1].
* Preserving outliers in full-precision also resembles existing methods such as LLMint8 [2].
* Preserving arbitrary weight parameters in FP would induce computation inefficiencies because of hardware-unfriendly structure [3].
* From the ablation results in Table 3, the effect of frequency-aware calibration appears to be marginal.

[1] Shao, Wenqi, et al. "OmniQuant: Omnidirectionally Calibrated Quantization for Large Language Models." The Twelfth International Conference on Learning Representations.

[2] Dettmers, Tim, et al. "Gpt3. int8 (): 8-bit matrix multiplication for transformers at scale." Advances in neural information processing systems 35 (2022): 30318-30332.

[3] Lin, Ji, et al. "Awq: Activation-aware weight quantization for on-device llm compression and acceleration." Proceedings of machine learning and systems 6 (2024): 87-100.

**Questions:**

* Does sampling-based quantization occur online during inference or are scaling factors pre-computed during calibration?
* If it occurs online, how does it affect on throughput?
In my understanding, strength of smoothing based approach is that the smoothing factor is fused into weight so that the inference cost remains the same.
Is outlier-aware quantizer better than other online outlier-handling approaches such as OCS even without fusing?
* Could you further review related work and clarify how the proposed outlier-aware quantizer differs from existing methods?
* Could you provide additional ablation results under more diverse settings, such as different datasets or bit-widths?
* Does applying and training multiple kernels not lead to higher training cost or longer training time?

---

> ### Author Response · Authors · 2025-11-24
> **Response to Reviewer piwj (denoted as R1) part 1**
>
> ``Q1-1`` Smoothing and clipping approach of outlier-aware quantizer strongly resembles existing methods such as OmniQuant.
>
> ``A1-1`` Thank you for the comment.
>
> Both the outlier-aware quantizer and OmniQuant follow the same high-level **smoothing-and-clipping scheme**. However, OmniQuant is designed to quantize large language models (**LLMs**), whereas the outlier-aware quantizer targets an image demoiréing model composed mainly of **convolutional modules**. Consequently, the two approaches differ substantially in their specific design choices.
>
> **(1) Different clipping targets.**  OmniQuant employs learnable **weight** clipping in LWC.   By contrast, in convolutional modules the weights are confined to small kernels and are relatively few per layer compared with the volume of activations. Directly **truncating weight outliers can therefore cause disproportionate accuracy loss**, as shown in the table below:
>
> |           method            |  PSNR   |  SSIM  | LPIPS  |
> | :-------------------------: | :-----: | :----: | :----: |
> | truncating weight outliers  | 16.2965 | 0.6724 | 0.4103 |
> | quantize weight with minmax | 20.5291 | 0.7538 | 0.3247 |
>
> Consequently, in the **outlier-aware quantizer** we clip only the **activation** quantization bounds rather than clipping the weights.
>
> **(2) Choice of learnable parameters.** OmniQuant treats both the **smoothing factor** and the weight **clipping bounds** as learnable. In our outlier-aware quantization, **only the activation clipping bound** is learnable; the smoothing factor is initialized using a sampling-based estimator and then **kept fixed**. Because the rounding operation in quantization can **destabilize training**, we observe that our design outperforms the alternative that makes both the smoothing factor and the weight clipping bounds learnable, as the results shown in the table (experiments are conducted in 4bit, UHDM dataset):
>
> |                         method                         |  PSNR   |  SSIM  | LPIPS  | training epoch |
> | :----------------------------------------------------: | :-----: | :----: | :----: | :------------: |
> |          Ours (only learn the clipping bound)          | 21.0808 | 0.7626 | 0.3068 |       4        |
> | Omniquant (learn clipping bound and smoothing factor ) | 21.0643 | 0.7613 | 0.3106 |       40       |
>
> Furthermore, by initializing the smoothing factor to a good value via sampling and reducing the number of learnable parameters, our method requires **only 4 training epochs**, whereas the configuration with learnable smoothing factor and clipping bounds typically requires **40 epochs**. Consequently, our approach substantially reduces the training time.
>
> ``Q1-2`` Preserving outliers in full-precision also resembles existing methods such as LLM.int8.
>
> ``A1-2`` Thank you for the comment.
>
> Indeed, many prior methods adopt mixed-precision quantization. For example, LLM.int8[1] processes a very small number of outlier feature dimensions in FP16, and SqueezeLLM[2] stores a small subset of weights in a sparse format. However, few approaches are designed specifically for **convolutions**. Because outlier weights in convolution kernels occur **sporadically**, existing coarse-grained mixed-precision schemes are not well suited, and directly converting convolutions into **sparse matrix multiplications introduces additional overhead**. To address this, we implement **a dedicated kernel for the FP16 branch** that parallelizes processing of FP16 weights at the element level.
>
> > [1] Dettmers, Tim et al. “LLM.int8(): 8-bit Matrix Multiplication for Transformers at Scale.” *ArXiv* abs/2208.07339 (2022): n. pag.
> >
> > [2] Kim, Sehoon et al. “SqueezeLLM: Dense-and-Sparse Quantization.” *ArXiv* abs/2306.07629 (2023): n. pag.
>
> ``Q1-3``  Preserving arbitrary weight parameters in FP would induce computation inefficiencies because of hardware-unfriendly structure.
>
> ``A1-3``  Thank you for the comment.
>
> The additional overhead introduced by the FP16 branch is a legitimate concern. We have implemented a kernel that processes outlier-weight computations **element-wise**. Owing to this targeted design and the scarcity of outliers, the FP16 path incurs only 7% additional runtime and 6% additional GPU memory.

---

> ### Author Response · Authors · 2025-11-24
> **Response to Reviewer piwj (denoted as R1) part 2**
>
> ``Q1-4`` From the ablation results in Table 3, the effect of frequency-aware calibration appears to be marginal.
>
> ``A1-4``  Thank you for the comment.
>
> Traditional image quality assessment (IQA) metrics do not accurately reflect the impact of banding artifacts[3].  Therefore, the advantage of  frequency-aware calibration in IQA metrics is not significant. To visually demonstrate the effectiveness of frequency-aware calibration, we present a comparison of the visual results from the model trained with original loss and the one incorporating our method **(see Figure 1 in the supplementary)**. It is evident that under the extremely low bit-width setting of 4-bit, the baseline model (using the original loss) still suffers from severe banding artifacts. In contrast, frequency-aware calibration significantly alleviates this issue.
>
> > [3] Y. Wang, S. -U. Kum, C. Chen and A. Kokaram, "A perceptual visibility metric for banding artifacts," 2016 IEEE International Conference on Image Processing (ICIP), Phoenix, AZ, USA, 2016, pp. 2067-2071, doi: 10.1109/ICIP.2016.7532722.
>
> ``Q1-5`` Does sampling-based quantization occur online during inference or are scaling factors pre-computed during calibration?
>
> ``A1-5`` Thank you for the comment.
>
> To reduce inference overhead, we employ **static quantization**, where the sampling-based quantization is performed during the calibration stage, and smoothing factors are **pre-computed**.
>
> ``Q1-6`` If it occurs online, how does it affect on throughput? In my understanding, strength of smoothing based approach is that the smoothing factor is fused into weight so that the inference cost remains the same. Is outlier-aware quantizer better than other online outlier-handling approaches such as OCS even without fusing?
>
> ``A1-6`` Thank you for the comment.
>
> Since sampling-based quantization is performed during the calibration stage， it does not affect the throughput.
>
> We evaluated our OCS method against an outlier-aware quantizer in terms of both model performance and inference speed. To ensure a fair comparison, both methods utilized 4-bit quantization, and the outlier-aware quantizer was implemented **without any training-based calibration**. In terms of performance,  the outlier-aware quantizer outperforms OCS, as OCS is limited to handling outliers within a single channel. Regarding inference overhead, the outlier-aware quantizer also achieved faster inference speeds, even without operator fusion, because OCS introduces the **additional operations of copying and concatenating activation channels**. The experimental results are as follows:
>
> |             METHOD             |         PSNR         |        SSIM         |       LPIPS        | Time/ms | MEMORY/MB |
> | :----------------------------: | :------------------: | :-----------------: | :----------------: | :-----: | :-------: |
> | Ours (outlier-aware quantizer) | 20.2803 $\pm 0.0021$ | 0.6745 $\pm 0.0005$ | 0.3369$\pm 0.0005$ |   160   |   1238    |
> |              OCS               |       16.6363        |       0.4255        |       0.6831       |   191   |   1631    |

---

> ### Author Response · Authors · 2025-11-24
> **Response to Reviewer piwj (denoted as R1) part 3**
>
> ``Q1-7`` Could you further review related work and clarify how the proposed outlier-aware quantizer differs from existing methods?
>
> ``A1-7``  Thank you for the comment.
>
>  Our outlier-aware quantizer, while sharing high-level concepts with prior work, is distinguished by its specific design for convolutional architectures.
>
> **(a) Comparison with Smoothing-and-Clipping Methods (e.g., OmniQuant):** While both methods use a smoothing-and-clipping approach, their designs differ significantly due to their target models (convolutional networks vs. LLMs).
>
> **Clipping Target:** OmniQuant clips model **weights**. In contrast, our method clips the **activation** quantization bounds. We found that directly clipping weights in small convolutional kernels can cause a disproportionate loss in accuracy.
>
> **Learnable Parameters:** OmniQuant learns **both the smoothing factor and weight clipping bounds**. Our method **only learns the activation clipping bound**, while the smoothing factor is initialized via a sampling-based estimator and then fixed. This design choice enhances training stability and reduces the required training from 40 epochs to just 4.
>
> **(b) Comparison with Mixed-Precision Methods (e.g., LLM.int8):** While methods like LLM.int8 also use mixed precision, they are **not optimized for convolutions** where outlier weights appear sporadically.
>
> **Specialized Kernel:** Existing coarse-grained, mixed-precision schemes are ill-suited for demoiréing models, and converting convolutions to sparse matrix multiplications introduces significant overhead. To address this, we developed a dedicated kernel that efficiently parallelizes the processing of FP16 outlier weights at the element level, adding negligible overhead due to the small proportion of outlier weights.
>
> ``Q1-8`` Does applying and training multiple kernels not lead to higher training cost or longer training time?
>
> ``A1-8``  Thank you for the comment.
>
> The additional training time introduced by the frequency extraction module is negligible, as it only incorporates a few extra convolutional operations. For comparison, the training times **with the original loss** and **with frequency extraction** were **108s** and **116s** in one epoch, respectively.

---

### Author Response · Authors · 2025-11-24
**Response to all reviewers and area chairs for a brief summary**

Dear reviewers and area chairs,

We are grateful to the reviewers and area chairs for their valuable time and thoughtful feedback. We provide the following summary of our responses.

We are pleased that:

1. All reviewers recognize the outstanding performance of our quantization method.
2. R1 acknowledges that our frequency-aware calibration is effective.
3. R2 and R3 emphasize the practicality of our method.

We have provided detailed responses to each reviewer and offer the following summary:

1. We present a detailed mathematical analysis of the **sampling method**’s performance and variance.
2. We execute our method repeatedly and compute the **variance**, demonstrating its **stability**.
3. We provide additional experiments demonstrating that our sampling method outperforms **fixed-percentile clipping.**
4. We demonstrate the necessity of **quantization boundary optimization** through experiments.
5. We analyze the effect of our **frequency-aware calibration** through visual results.
6. We detail the implementation of **mixed-precision** weights and the additional overhead.
7. We report the full **experimental settings** used in all ablation studies.
8. We compare our method with **related quantization approaches** and clarify our contributions.
9. We explain the meaning of **OPS** and justify the completeness of its computation.

We again thank all reviewers and area chairs.

Best regards,

Authors

---

### Comment · Area_Chair_p9qe · 2025-11-28

Dear Reviewers,

Thank you for your valuable time and expertise in reviewing this paper.

The authors have now submitted their rebuttal. We would appreciate it if you could review their responses and assess whether your concerns have been addressed.

Best regards,

AC

---

### Author Response · Authors · 2025-11-30
**Rebuttal-Discussion Summary and Reviewer Feedback Status**

Dear Program Chairs, Senior Area Chairs, Area Chairs, and Reviewers:

We would like to express our deep appreciation to the program chairs, senior area chairs, area chairs, and all reviewers (R1-piwj, R2-FcUX, R3-H9oE) for their valuable time and expertise in reviewing this paper.

To support the decision-making process, we present below a brief overview of our rebuttal responses along with an update on the current reviewer feedback status.

### Summary of Discussion During Rebuttal

Overall, a majority of the reviewers have recognized the primary contributions of our submission, such as:

1. Our work achieves the **state-of-the-art performance**.
2. The **frequency-aware calibration** is simple and effective.
3. Our method demonstrates significant **practical applicability**.

Within the discussion period, we addressed every question and concern presented, including:

1. We clarify the novelty and contributions of our **outlier-aware quantizer** through additional comparisons with existing methods(Omniquant,  LLMint8, SqueezeLLM) and theoretical analysis.
2. More **experiments**: comparisons to Percentile, OCS and Omniquant, more ablation studies,
3. **Overhead** evaluation of FP16 mixed-precision kernel.
4. Additional **visual comparisons** to demonstrate the effectiveness of **Frequency-aware calibration**.

### Summary of Reviewer Feedback Status

1. **R1-piwj**: We have provided comprehensive explanations and experimental evidence addressing all concerns raised by this reviewer. We believe these additions fully resolve the issues. Since the reviewer did not engage in the discussion phase, we believe the initial assessment may not account for our substantial revisions and respectfully request reconsideration based on the updated submission.
2. **R2-FcUX:** The reviewer acknowledged the effectiveness of our sampling-based approach while questioning certain aspects of novelty. We clarified these points through detailed explanations consistent with our initial response. The reviewer **did not raise additional concerns **following our rebuttal, suggesting that the main issues have been addressed.
3. **R3-H9oE:** The reviewer raised multiple technical questions. We have provided detailed clarifications, corrected misunderstandings about our approach, and supplied additional experimental evidence for each point. The reviewer **acknowledged the potential** of our FP16-branch implementation, and we believe our comprehensive responses have effectively addressed their concerns.

### Overall

Our paper makes contributions in outlier-aware quantization, frequency-aware calibration, and practical deployment efficiency. Except for reviewer piwj, who did not take part in the discussion, the reviewers' views converge toward a favorable consensus.



We sincerely hope that the final decision may take these observations into careful consideration.

Thank you again for your time, constructive feedback, and support.

Best regards,

Authors

---

### Meta-Review · Area_Chair_Mj86 · 2025-12-09

**Summary:**

This paper proposes a PTQ framework for demoiréing, with its main contributions being the use of FP16 to reduce outlier errors and the introduction of frequency-domain space to alleviate banding artifacts. The paper received three critical reviews (422). The reviewers raised questions regarding the paper's novelty and motivation, and noted a lack of some experimental results and important descriptions. While these issues were partially addressed in the rebuttal, they are difficult to be fully resolved within a single round of revisions. Therefore, the paper is rejected.

**Reviewer Concerns:**

While the authors have partially addressed the reviewers' concerns regarding the motivation and experimental design, the core issue of presentation quality and insufficient novelty remain the fundamental limitations.

**Reviewer Scores:**

Even with full discussion, the overall presentation quality and methodological novelty remain below the acceptance threshold.

---

### Decision · Program_Chairs · 2026-01-26

Reject